# Mechanism of fertilization-induced auxin synthesis in the endosperm for seed and fruit development

Lei Guo [1], Xi Luo [1], Muzi Li[1], Dirk Joldersma[1], Madison Plunkert [1] & Zhongchi Liu [1]✉

The dominance of flowering plants on earth is owed largely to the evolution of maternal tissues such as fruit and seedcoat that protect and disseminate the seeds. The mechanism of how fertilization triggers the development of these specialized maternal tissues is not well understood. A key event is the induction of auxin synthesis in the endosperm, and the mobile auxin subsequently stimulates seedcoat and fruit development. However, the regulatory mechanism of auxin synthesis in the endosperm remains unknown. Here, we show that a type I MADS box gene *AGL62* is required for the activation of auxin synthesis in the endosperm in both *Fragaria vesca*, a diploid strawberry, and in Arabidopsis. Several strawberry *FveATHB* genes were identified as downstream targets of *FveAGL62* and act to repress auxin biosynthesis. In this work, we identify a key mechanism for auxin induction to mediate fertilization success, a finding broadly relevant to flowering plants.

[1] Department of Cell Biology and Molecular Genetics, University of Maryland, College Park, MD 20742, USA. ✉email: zliu@umd.edu

The dominance of flowering plants on earth owes much to the evolution of maternal tissues such as seedcoat and fruit that protect and disseminate the seeds. However, how fertilization induces the development and differentiation of these maternal tissues is not well understood. Strawberry has a history of serving as a model to study how fertilization induces fruit development due to its unique flower and fruit structure. The fleshy fruit develops from the receptacle, an enlarged stem tip supporting hundreds of seed-containing ovaries (achenes)[1,2]. The achenes dotting the surface of the receptacle are easily accessible for dissection and manipulation, and their individual impact on the underlying receptacle fruit flesh development is visible in real time as they develop. When achenes were removed from the receptacle, no fruit flesh would form; however exogenous application of auxin promoted receptacle fruit enlargement in the absence of achenes[3]. Therefore, auxin was identified as an inductive signal produced by the seed to stimulate fruit development[3–5]. The essential role of auxin in fruit induction is supported by genetic studies in plants such as Arabidopsis and tomato, where mutants with defects in auxin signaling develop virgin (or parthenocarpic) fruits[6–11], suggesting a conserved auxin-dependent mechanism in flowering plants. Genome-wide transcript profiling and reporter gene expression in diploid strawberry, *Fragaria vesca*, revealed that the expression of auxin biosynthesis genes, *YUCs* and *TAAs*, are primarily induced in the endosperm immediately post-fertilization[5,12]. However, the molecular mechanism of how fertilization induces auxin synthesis in the endosperm for fruit induction is presently unknown.

In seeds, the embryo and endosperm are the products of double fertilization of the egg cell and central cells, respectively. While the endosperm is largely thought of as a nutritive tissue for the embryo, increasing evidence suggests the endosperm serves as a fertilization 'sensor' that coordinates the development of the embryo and the surrounding maternal sporophytic tissues such as seedcoat and fruit[13]. In Arabidopsis, transgenic expression of auxin biosynthesis genes, *DD65::TAA1* and *DD65::YUC6*, in the central cell initiates parthenocarpic endosperm proliferation, seedcoat differentiation, and fruit development[14,15]. These studies not only established auxin as the fertilization-induced signal but also pinpointed endosperm as the tissue that senses fertilization, produces auxin, and exports auxin to the seedcoat and fruit.

In most angiosperms, endosperm development initiates as a syncytium in which the fertilized central cell nucleus divides several times without cytokinesis. Sometime later, the timing varying between species, the endosperm nuclei become cellularized[16,17]. Importantly, the timing of endosperm cellularization determines the extent of endosperm nuclear proliferation and hence seed size and weight. In Arabidopsis, fertilization-induced auxin production is essential for endosperm nuclear proliferation[14]. Increased or prolonged auxin biosynthesis in the endosperm prevents endosperm cellularization and leads to seed arrest, but reduced auxin biosynthesis can restore cellularization, suggesting that the auxin level in the endosperm may determine the timing of endosperm cellularization[18]. Thus, understanding the regulation of auxin biosynthesis in endosperm has significant impact on grain and fruit yields.

In Arabidopsis, *AtAGL62*, a type I MADS-box gene, was identified as a regulator of endosperm cellularization. *AtAGL62* is expressed exclusively in early endosperm development, during the nuclear proliferation phase, and its expression declines quickly before cellularization[19,20]. While *atagl62* loss-of-function mutants showed precocious endosperm cellularization, ectopic *AtAGL62* expression blocked endosperm cellularization[19,21,22]. As such, the effect of *AtAGL62* mirrors that of auxin in promoting endosperm nuclear proliferation and inhibiting endosperm cellularization. However, the functional relationship

between *AtAGL62* and auxin biosynthesis remains unknown. A prior study suggested that *AtAGL62* positively regulates the expression of an ABCB-type transporter that is required for auxin transport from endosperm to the seedcoat[15,20]. This observation does not, however, explain the similar effects of *AtAGL62* and auxin in endosperm cellularization and development.

In this work, we investigated the mechanisms of how fertilization induces auxin biosynthesis in the endosperm to promote fleshy fruit initiation using diploid wild strawberry *Fragaria vesca* as our model. We showed that *FveAGL62* promotes fruit development by promoting auxin biosynthesis in the endosperm. Further, we demonstrated that Arabidopsis *AtAGL62* similarly regulates auxin synthesis in the endosperm. Thus, our investigation has uncovered an evolutionarily conserved mechanism underlying fertilization-induced auxin synthesis in the endosperm. Our finding is broadly relevant to flowering plants for producing seeds and fruits and provides a knowledge base for increasing grain and fruit size as well as creating virgin (parthenocarpic) fruits.

## Results

**FveAGL62 is essential for seed and fruit development in *F. vesca*.** To identify key regulators that underlie fertilization-induced auxin biosynthesis in seeds, we mined prior *F. vesca* co-expression networks based on transcriptome data from distinct flower and fruit tissues at different stages[5,23,24]. A consensus co-expression cluster was identified with its eigengene highly and preferentially expressed in the stage 2 seeds (Supplementary Fig. 1a; Supplementary Data 1). Stage 2 seeds are formed at 2-4 days post anthesis (DPA)[1] and contain transcripts induced by fertilization. Among the transcription factors in the cluster (Supplementary Fig. 1b), two type I MADS-box genes *FvH4_2g03030* and *FvH4_6g08460* exhibit highly specific and transient expression in the ghost (endosperm and seed coat) (Supplementary Fig. 1d, e) and are chosen for further characterization.

First, phylogenetic analysis of 45 *F. vesca* and 74 Arabidopsis type-I MADS-box genes (Supplementary Fig. 1c) indicates that *FvH4_2g03030* clusters with *AtAGL62* and *FvH4_6g08460* clusters with *AtAGL80* and are thereafter referred to as *FveAGL62* and *FveAGL80* respectively. Next, to verify their expression, transgenic *F. vesca* plants containing reporter genes *ProFveAGL62::FveAGL62-GUS* and *ProFveAGL80::FveAGL80-GUS* were generated. In each case, GUS expression was detected in the endosperm at stage 2 (2-4 DPA) but not in the pre-fertilization stage 1 ovule (Fig. 1a–c), confirming that *FveAGL62* and *FveAGL80* are specifically expressed in the endosperm soon after fertilization.

CRISPR/Cas9 was used to knockout *FveAGL62*. Two sgRNA (sgRNA1 and sgRNA3) targeting different regions of the *FveAGL62* ORF were independently cloned and transformed into *F. vesca*. Transgenic plants containing the Cas9-sgRNA1 led to three independent mutant lines, which carried either homozygous or biallelic frameshift mutations (Supplementary Fig. 2a). Transgenic plants containing the Cas9-sgRNA3 yielded three additional lines, all of which contained heterozygous frameshift mutations (Supplementary Fig. 2b). The three homozygous or biallelic *fveagl62* mutant lines, *fveagl62*$^{-1/-4}$, *fveagl62*$^{-1/-1}$, and *fveagl62*$^{-1/-5}$, showed similar fruit phenotypes. Specifically, they were morphologically similar to wild type during vegetative growth (Fig. 1d), but their fruits did not develop further beyond stage 2 (Fig. 1e; Supplementary Fig. 3a). While *fveagl62* ovules at stage 1 showed no discernable difference from those of wild type, *fveagl62* seeds at stage 2 were much smaller than wild type, appeared to dry up, and were not viable (Fig. 1f).

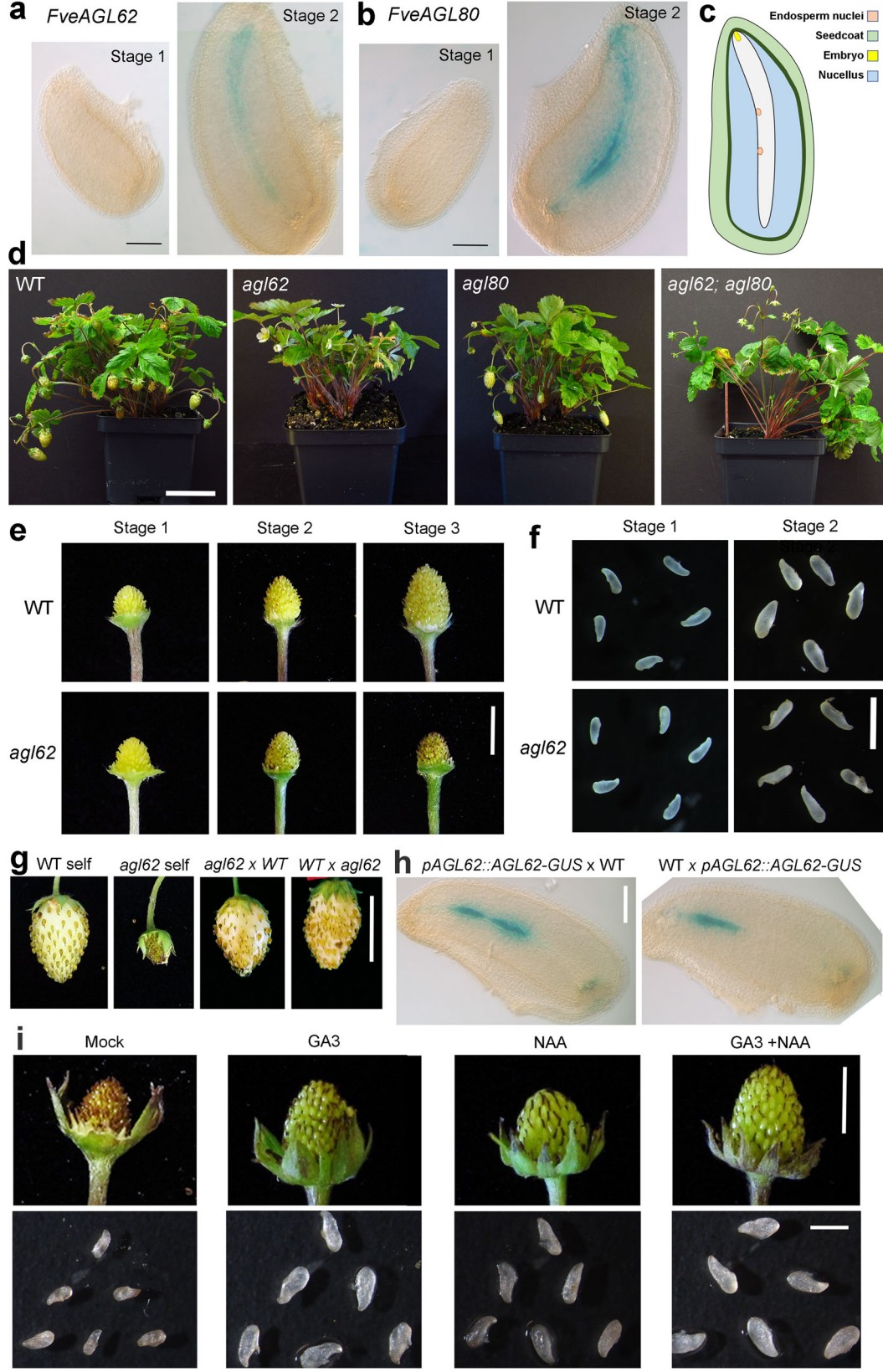

CRISPR/Cas9 was also used to generate a *fveagl80⁻¹/⁻¹* mutant (Supplementary Fig. 2c), which exhibited normal vegetative growth and developed fruit similarly to wild type (Fig. 1d). Since *FveAGL80* (*FvH4_6g08460*) clusters together with four additional *FveAGL80-Like* (*FveAGL80L*) genes in *F. vesca* (Supplementary Fig. 1c), functional redundancy may have prevented any mutant phenotype of *fveagl80⁻¹/⁻¹*. Finally, *fveagl62⁻¹/⁻⁴*, *fveagl80⁻¹/⁻¹* double mutants exhibited a similar phenotype as the *fveagl62⁻¹/⁻⁴* single mutant (Fig. 1d).

MADS-box proteins often form heterodimers or tetramers in executing their functions, so we investigated whether FveAGL62 and FveAGL80 act as heterodimers like their Arabidopsis

**Fig. 1 Characterization of *F. vesca* mutants of *FveAGL62* and *FveAGL80*. a** *ProFveAGL62::FveAGL62-GUS* reporter expression in stage 1 ovule (pre-fertilization) and stage 2 seed (post-fertilization). **b** *ProFveAGL80::FveAGL80-GUS* reporter expression in stage 1 ovule and stage 2 seed. **c** Diagram of a stage 2 strawberry seed. **d** Mature *F. vesca* plants of WT, *fveagl62* single*, fveagl80* single, and *fveagl62 fveagl80* double mutant. *fveagl62* single and *fveagl62 fveagl80* double mutants fail to produce fruits. **e** Comparison of receptacle fruit development at the three earliest stages of fruit development. **f** WT and *fveagl62* stage 1 ovules and stage 2 seeds. **g** Fruits from self-cross or reciprocal crosses between WT and *fveagl62* (female genotype is listed first). Only self-cross of *fveagl62* leads to failed fruit growth. **h** GUS expression in stage 2 seeds derived from reciprocal crosses between *ProFveAGL62::FveAGL62-GUS* transgenic plants and WT. **i** *fveagl62* mutant fruits and seeds after different hormone treatments. Scale bars: 0.12 mm (**a**, **b**, **h**), 5 cm (**d**), 0.5 cm (**e**), 1 mm (**f**), 1 cm (**g**), 0.5 cm (fruit) (**i**) and 1 mm (seed) (**i**).

homologs[19,25]. Both yeast two-hybrid (Y2H) and bimolecular fluorescence complementation (BiFC) assays showed that FveAGL62 and FveAGL80 interacted with each other but neither could homodimerize (Supplementary Fig. 2d, e). BiFC was also used to test any interaction between FveAGL62 and the other four FveAGL80Ls. Positive interactions were observed between FveAGL62 and all four FveAGL80Ls, but no interaction was found between FveAGL80 and FveAGL80Ls (Supplementary Fig. 4). The data suggest that FveAGL62 can form heterodimers with FveAGL80 as well as FveAGL80L1-4.

**Characterization of *fveagl62* mutants in strawberry**. To determine whether *FveAGL62* is subject to parental imprinting, reciprocal crosses between *fveagl62* and wild type *F. vesca* were performed. The resulting F1 seeds and fruits were normal whether the *fveagl62* mutant served as the female or male donor (Fig. 1g). Only when both parents carried the *fveagl62* mutation, did the F1 progeny exhibit the mutant phenotype as shown in a self-cross of the *fveagl62* mutant (Fig. 1g). Reciprocal crosses were performed between a *ProFveAGL62::FveAGL62-GUS* transgenic plant and a non-transgenic WT plant; the *GUS* reporter expression was detected in the endosperm nuclei of young F1 seeds whether the *ProFveAGL62::FveAGL62-GUS* transgenic plant served as the male or female parent (Fig. 1h). Therefore, *FveAGL62* is not subjected to parental imprinting, which is similar to the Arabidopsis *AtAGL62*[19].

We then asked whether the *fveagl62* seed and fruit defects were caused by failures of auxin and GA biosynthesis. GA3 and auxin (NAA), alone or in combination, were applied to fertilized *fveagl62* mutant flowers and shown to cause fruit and seed enlargement, with the combined GA3/NAA treatment having the strongest effect (Fig. 1i). However, hormone treated *fveagl62* seeds eventually died, indicating that *fveagl62* may cause additional defects other than hormone production. Nevertheless, this result suggested that a loss of *FveAGL62* might result in reduced auxin and GA production and/or transport.

**RNA-seq profiling of *fveagl62* stage 2 seeds**. To identify downstream genes and processes regulated by *FveAGL62* during seed development, we used RNA-seq to profile stage 2 *fveagl62*$^{-1/-4}$ seeds and wild type seeds, each with three biological replicates. Differential gene expression (DE) analysis showed that 1102 and 1572 genes are respectively down- and up- regulated in the *fveagl62*$^{-1/-4}$ seeds (Supplementary Data 2). Among the down-regulated genes in *fveagl62*, enriched GO terms include cell wall modification and organization processes and seed development (Supplementary Fig. 5a; Supplementary Data 3). In contrast, enriched GO terms for up-regulated genes in *fveagl62* include responses to biotic stimuli and response to abscisic acid (Supplementary Fig. 5b; Supplementary Data 3).

Among all 30 type I MADS-box genes expressed in the WT or *fveagl62* stage 2 seeds (Supplementary Fig. 5c), *FveAGL80*, *FveAGL80L1,* and *FveAGL80L2* were also significantly down-regulated in the *fveagl62* seeds. Hence, *FveAGL62* may positively

regulate *FveAGL80* and *FveAGL80L* expression. The existence of these similarly regulated *FveAGL80/FveAGL80L* genes also explains a lack of phenotype of *fveagl80*$^{-1/-1}$ single mutants.

Most significantly, the RNA-seq data showed that the expression of several auxin biosynthesis genes, *FveYUC1*, *FveYUC5*, *FveYUC10*, *FveTAA1*, *FveTAR1*, and *FveTAR2*, are reduced in the *fveagl62* stage 2 seeds (Fig. 2a). RT-qPCR confirmed reduced transcript levels of five auxin biosynthesis genes in the *fveagl62* stage 2 seeds when compared with wild type (Fig. 2c). Similarly, the GA biosynthesis genes *FveGA20OX1c*, *FveGA20OX1d*, *FveGA3OX1a*, and *FveGA3OX1b* also showed reduced expression in *fveagl62* mutant seeds (Fig. 2b). Together, a loss of *FveAGL62* caused significant reduction in the transcript level of auxin and GA biosynthesis genes in stage 2 seeds.

We subsequently examined a prior *F. vesca* RNA-seq dataset[26] for the temporal expression pattern of these auxin and GA biosynthesis genes in seeds. We found that the transcripts of many auxin biosynthesis genes, *FveYUC5* and *FveYUC10*, *FveTAA1*, and *FveTAR1-3* (Supplementary Fig. 6a), and GA biosynthesis genes, *FveGA3OX1a, 1b, 1c, 1d* and *FveGA20OX1d* (Supplementary Fig. 6b), exhibited a gradual increase from stage 1 ovules to stage 3 seeds. RT-qPCR was performed for five auxin biosynthesis genes and four GA biosynthesis genes (Supplementary Fig. 6c), all of which showed the same expression trend from a low expression at stage 1 ovules to a high expression at stage 3 ghost (endosperm and seedcoat), consistent with their roles in fertilization-induced auxin and GA biosynthesis in seeds.

**Characterization of *fveagl62* endosperm phenotype**. To determine whether the reduced transcript level of auxin biosynthesis genes leads to reduced auxin activity in the *fveagl62* seeds, an auxin reporter *DR5ver2::GUS*[12,27] was introduced into the *fveagl62* mutant by genetic crosses. Compared with the wild type, *DR5ver2::GUS* expression is greatly reduced in the *fveagl62* stage 2 seeds (Fig. 2d, e), although the expression pattern is similar with more intense staining at the chalazal end of the endosperm and seed coat, which was proposed as the auxin transport route to the receptacle[12]. The result established that the *fveagl62* mutant seeds contain a lower auxin activity, which likely contributes to defects in the *fveagl62* seed and fruit development.

We used confocal laser scanning microscopy (CLSM) to examine *fveagl62* seeds in detail. Strawberry endosperm initially consists of a thin and elongated tube (Fig. 1c). The wild type stage 2 endosperm consists of several nuclei situated at the peripheral of the tube. In contrast, all three *fveagl62* mutant lines showed premature endosperm cellularization when the seeds were examined at stage 2 (Fig. 2f; Supplementary Fig. 3b). Both the confocal image and the confocal stack video (Supplementary movies 1 and 2) clearly illustrate the tube-like endosperm filled with a single file of cells separated by cell walls. Hence, similar to Arabidopsis *atagl62* mutants[19], strawberry *fveagl62* also exhibits premature endosperm cellularization.

To eliminate the possibility that the observed early cellularization phenotype was due to off-target mutations, we also examined *fveagl62* mutant seeds derived from an independent CRISPR

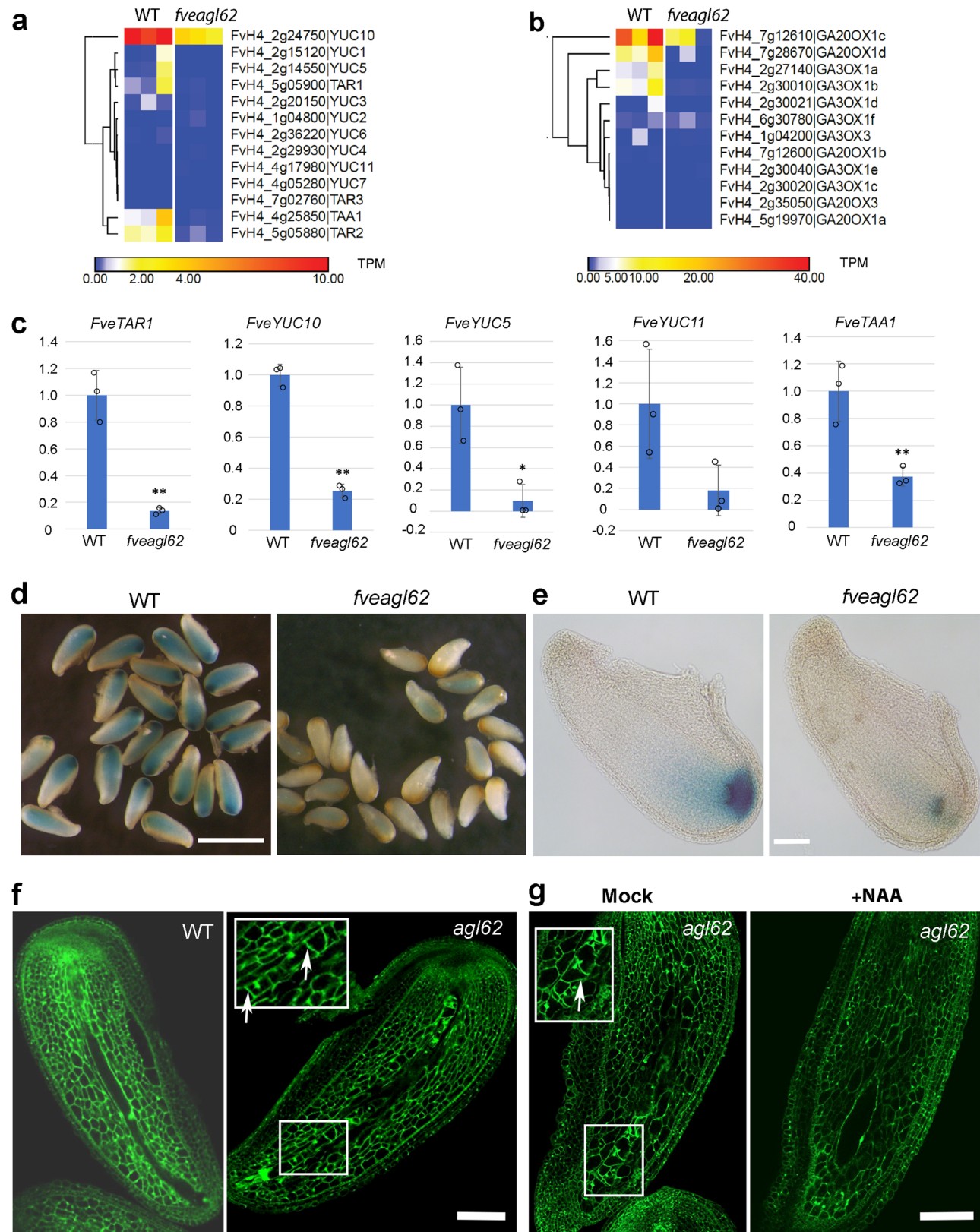

experiment, where a Cas9-sgRNA3 construct targeting a different sequence of *FveAGL62* was created and transformed into *F. vesca*. Three transgenic lines respectively with a −1, −1, or −4 bp mutation were obtained (Supplementary Fig. 2b). Since they are all heterozygous, the self-fertilized seeds of line #1 with a −1 bp mutation were examined under the confocal microscope. 25.9%

of the F1 seeds ($n = 54$) exhibited early cellularization of the endosperm (Supplementary Fig. 3c), supporting that the early endosperm cellularization was due to a defective *FveAGL62*.

We then tested if reduced auxin biosynthesis in *fveagl62* mutant seed may cause early endosperm cellularization. We applied auxin (NAA) to *fveagl62*$^{-1/-4}$ seeds at 1 DPA to test if

**Fig. 2 *F. vesca fveagl62* mutant seeds show reduced auxin biosynthesis. a** Hierarchical clustering of auxin biosynthesis gene expression in WT and *fveagl62* stage 2 seeds. **b** Hierarchical clustering of GA biosynthesis gene expression in WT and *fveagl62* stage 2 seeds. **c** RT-qPCR analysis of five auxin biosynthesis genes in WT and *fveagl62* stage 2 seeds. Gene IDs are indicated in **a**. Y-axis indicates the relative expression level against the control gene *FvePP2a* (FvH4_4g27700). Significant difference by two-tailed Student's *t*-test is indicated by ** ($P < 0.01$) or * ($P < 0.05$) between WT and *fveagl62* mutant seed. Error bars indicate standard deviation. Similar results were obtained in three biologically independent experiments. **d** *DR5ver2::GUS* reporter expression in WT and *fveagl62* stage 2 seeds. **e** *DR5ver2::GUS* expression in individual stage 2 seeds. **f** Confocal laser scanning microscopy images of WT and *fveagl62* seeds at stage 2. Precocious cellularization of *fveagl62* endosperm (arrows) is observed in stage 2 seeds. **g** Confocal laser scanning microscopic images of WT and *fveagl62* stage 2 seeds mock- or NAA-treated. Scale bars: 1 mm (**d**), 120 µm (**e**), 100 µm (**f**–**g**).

the NAA application could reduce premature endosperm cellularization. 11.9% of *fveagl62*$^{-1/-4}$ seeds ($n = 42$) were rescued by the NAA treatment, showing a lack of premature endosperm cellularization and enlarged nucellus 2 days after NAA treatment (Fig. 2g; Supplementary movie 4). Rescue was not observed in mock-treated *fveagl62* seeds ($n = 56$) (Fig. 2g; Supplementary movie 3). Therefore, reduced auxin in *fveagl62* seeds likely contributes to the premature endosperm cellularization, and auxin may act downstream of *FveAGL*, consistent with a positive role of *FveAGL62* for auxin biosynthesis gene expression (Fig. 2a, c).

**Characterization of *atagl62* mutants in Arabidopsis.** In Arabidopsis, despite a similar requirement of auxin and *AtAGL62* for endosperm nuclear proliferation[14,19], functional relationships between *AtAGL62* and auxin synthesis are unknown, although *AtAGL62* has been shown to be required for the expression of *P-GLYCOPROTEIN 10* (*PGP10*) that may transport auxin from the endosperm to the seedcoat[15]. Based on the similar endosperm phenotype observed in Arabidopsis *atagl62* seeds and strawberry *fveagl62* seeds, we hypothesized that *AtAGL62* may play a similar positive regulatory role for auxin biosynthesis as the strawberry *FveAGL62*. To test this hypothesis, we crossed an established Arabidopsis line containing the *R2D2* auxin reporter into an Arabidopsis *atagl62-2* mutant plant. The *R2D2* reporter construct contains *DII-VENUS* and *mDII-Tdtomato* reporter genes. While the DII-VENUS is subjected to auxin-mediated protein degradation, mDII-Tdtomato is not and serves as a control[27]. As shown in Fig. 3a, the DII-VENUS fluorescence signal is absent in the WT endosperm indicating auxin accumulation there. In contrast, strong nuclear DII-VENUS fluorescence is observed in *atagl62-2* endosperm nuclei (Fig. 3a) suggesting an absence of auxin there. Furthermore, the ratio of DII-VENUS fluorescence to mDII-Tdtomato fluorescence is significantly higher in the *atagl62-2* endosperm nuclei than the wild type (Fig. 3a), indicating an absence or severely reduced auxin accumulation in the *atagl62-2* endosperm.

The decreased auxin level in the *atagl62-2* endosperm may be caused by reduced auxin biosynthesis gene expression. A *ProAtYUC10::3xnGFP* transgene[28] was introduced into *atagl62-2* plants by genetic crosses to establish an F2 plant homozygous for the transgene and heterozygous for the *atagl62-2* mutation. In F3, 95.4% ($n = 121$) of the wild type seeds show strong GFP fluorescence in the endosperm nuclei (Fig. 3b); in contrast, 92.86% ($n = 56$) of *atagl62-2* ($-/-$) seeds showed undetectable or extremely weak GFP fluorescence in the endosperm nuclei (Fig. 3b). Together, the data revealed a significant reduction of auxin accumulation as well as reduced biosynthesis gene expression in the *atagl62* mutant endosperm, supporting a similar function of *AtAGL62* to *FveAGL62* in promoting auxin biosynthesis.

If the premature cellularization of *atagl62-2* endosperm is caused by reduced auxin, exogenous application of auxin may be able to rescue this endosperm defect. Fertilized Arabidopsis *atagl62-2* ($-/+$) siliques at 1 DPA were treated with a synthetic

auxin (2,4-dichlorophenoxyacetic acid; 2,4-D). We chose 2,4-D due to its efficacy in Arabidopsis seeds[14]. Progeny seeds at 2 DPA were examined under the confocal microscope. Different endosperm cellularization phenotypes including no cellularization, partial cellularization, and complete cellularization were quantified (Fig. 3c, d). We observed 22.7% (138 out of 608) mock-treated seeds exhibiting complete cellularization, a proportion that closely matches the 25% *atagl62*-2 ($-/-$) seeds expected from the *atagl62*-2 ($-/+$) parents. A small percentage (4.61%) of mock-treated seeds also showed partial endosperm cellularization (Fig. 3c, d). Upon 2,4-D treatment, only 15% (95 out of 633) seeds exhibited complete cellularization (Fig. 3d). If one focuses on the homozygous *atagl62-2* seeds, the proportion that exhibits complete cellularization would be reduced from 91 to 60% (22.7 and 15% divided by 25% respectively) by the auxin treatment. Hence, 2,4-D is able to partially rescue the endosperm defect of *atagl62-2* seeds.

Since strawberry *fveagl62* and Arabidopsis *atagl62* mutants both exhibit early endosperm cellularization and reduced auxin biosynthesis and accumulation, the strawberry *FveAGL62* and Arabidopsis *AtAGL62* appear to possess a conserved function. In support of this, the strawberry *FveAGL62* ORF flanked by the *AtAGL62* promoter and terminator was transformed into *atagl62-2*. Three independent transgenic lines were characterized in depth; the *FveAGL62* transgene is able to rescue all defects of *atagl62-2* including early endosperm cellularization and seedcoat development (Fig. 3e; Supplementary Fig. 7a–c).

**Seedcoat development in *fveagl62* mutants.** In Arabidopsis, *atagl62* mutant seedcoat fails to differentiate indicated by an absence of proanthocyanidins in the endothelium layer (Supplementary Fig. 7c)[15]. Hence, we investigated whether *FveAGL62* is also required for seedcoat development in strawberry. Stage 1 ovule and stage 2-3 seeds from *fveagl62*$^{-1/-4}$ strawberry mutants were dissected out of the ovaries and stained with vanillin that stains proanthocyanidins[29]. At stage 1, both wild type and *fveagl62* ovules showed no vanillin staining except mild staining at the two poles (Fig. 4a). At stage 2 however, strong vanillin staining was present in both wild type and *fveagl62* seedcoat. When the *fveagl62* mutant seeds arrested development at stage 3, they still stained dark red similarly to the wild type (Fig. 4a). CLSM optical sections revealed similar seed coat cell layers in wild type and *fveagl62* with three integument cell layers exterior to the endothelium layer (Fig. 4b). Hence, unlike Arabidopsis, the strawberry *FveAGL62* is not required for seedcoat development. Interestingly, *FveAGL62* ORF driven by the cis-elements of *AtAGL62* readily rescued the seedcoat defect of Arabidopsis *atagl62* mutants (Supplementary Fig. 7c), suggesting that the distinct requirement of *AtAGL62* for Arabidopsis seedcoat development is not due to different *AGL62* protein function but rather differences afforded by different reproductive structures or processes.

***FveATHBs* may mediate the effect of *FveAGL62* on auxin synthesis.** The above results established *FveAGL62* a positive

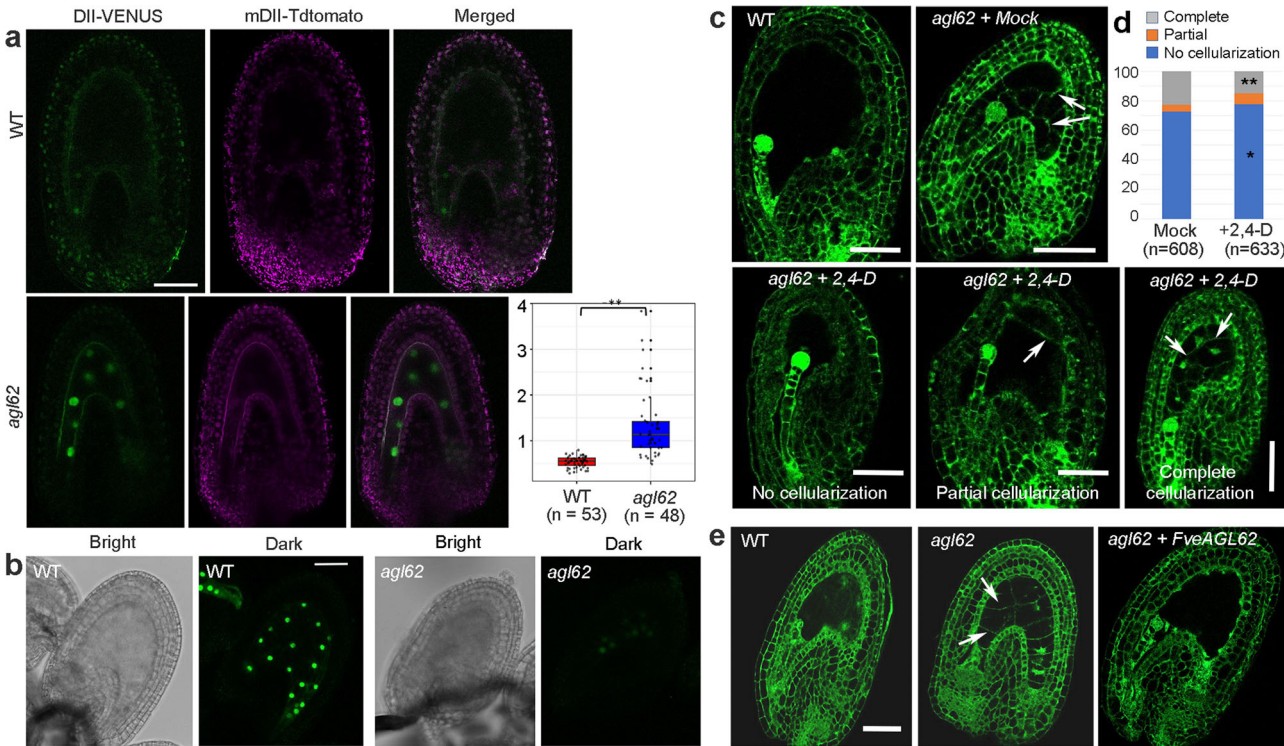

**Fig. 3 Arabidopsis atagl62 endosperms have reduced auxin when compared with WT. a** Confocal image of auxin reporter R2D2 in WT (Col-0) and *atagl62-2* seeds at 2 DPA. Absence of DII-VENUS nuclear signal in the WT endosperm contrasts with strong DII-VENUS nuclear signals in the *atagl62-2* endosperm. The scatter graph Y-axis indicates the ratio between DII-VENUS and mDII-Tdtomato signal per endosperm nucleus. A second independent experiment gave a silimar result. The elements of boxplots and source data are provided in the Source Data file. Significant difference indicated by ** ($P = 7.55e-12$ by two-tailed Student's *t*-test) is found between WT and *atagl62*. **b** Confocal image of *ProAtYUC10::3xnGFP* signal in WT and *atagl62* endosperms. Note the strong nuclear GFP signals in the WT endosperm and almost undetectable GFP signals in the *atagl62* endosperm. **c** Confocal image of WT and *atagl62* mutant seeds at 2 DPA either mock-treated or auxin (2,4-D)-treated. Different extent of endosperm cellularization is observed. Arrows indicate cell walls between endosperm cells. **d** Quantification of mock- or 2,4-D-treated seeds derived of *atagl62-2* (−/+) parents. Y-axis is the percentage of endosperms with one of the three phenotypes: no cellularization, partial cellularization, and complete cellularization. Significant difference (two-tailed Fisher's exact test) is indicated by ** ($P = 0.0006$) for complete cellularization or * ($P = 0.0416$) for no cellularization between mock and 2,4-D treatments of each phenotypic category. **e** Confocal image of Arabidopsis seeds at 2 DPA. Precocious endoserpm cellularization is absent in transgenic *atagl62* plants containing the strawberry *FveAGL62* gene. Scale bar in **a**–**e**: 50 μm.

regulator of auxin biosynthesis in the endosperm. Therefore, we examined if this positive effect of *FveAGL62* on auxin biosynthesis is direct or indirect. A yeast one-hybrid (Y1H) screen using the promoter of an auxin biosynthesis gene *FveYUC10* failed to identify FveAGL62 or FveAGL80 as binding factors. We then used Y1H assay to directly test if FveAGL62 and FveAGL80 could bind and activate auxin biosynthetic genes *FveYUC10* and *FveTAR1*. No activation of *FveYUC10* or *FveTAR1* was observed even when both *FveAGL62* and *FveAGL80* were simultaneously introduced into the yeast (Supplementary Fig. 8a). Therefore, *FveAGL62/FveAGL80* likely promotes the expression of auxin biosynthesis genes indirectly by regulating other transcription factors.

We re-examined the RNA-seq data from stage 2 *fveagl62* and WT seeds described above and noticed a subfamily of *ATHB* genes that were significantly up-regulated in *fveagl62* (Fig. 5a). This ATHB subfamily of transcription factors is composed of both zinc finger and homeobox domains, and one subfamily member was previously found to bind the *FveYUC10* promoter in a Y1H screen (Supplementary Fig. 8b). *FveATHB29a*, *FveATHB29b*, and *FveATHB30* not only exhibited an increased expression in the *fveagl62* seeds (Fig. 5a) but also showed a decreasing expression trend from stage 1 ovules, stage 2 seeds, to stage3 ghosts (endosperm and seedcoat) in the wild type (Fig. 5b). This expression trend is consistent with the hypothesis that fertilization-induced FveAGL62/FveAGL80 represses *FveATHB* gene

expression to relieve the inhibition of auxin biosynthesis gene expression during early seed development.

To test if the FveAGL62/FveAGL80 directly represses the expression of these *FveATHBs*, we conducted Y1H assay and transient luciferase (LUC) assays in tobacco leaves. In yeast, either FveAGL62 or FveAGL80 failed to activate the promoters of *FveATHB29b* and *FveATHB30*; when expressed together however, FveAGL62 and FveAGL80 were able to activate both *FveATHB* genes (Fig. 5c, d). In tobacco leaves, the LUC reporter driven by either *FveATHB29b* or *FveATHB30* promoter showed reduced expression when either *FveAGL80* alone or *FveAGL62* and *FveAGL80* together were introduced into the tobacco leaves (Fig. 5e, f), supporting a direct repressive role of *FveAGL62/FveAGL80* on the expression of *FveATHB29b* and *FveATHB30*.

As shown earlier, many auxin biosynthesis genes *FveYUC5*, *FveYUC10*, *FveYUC11*, *FveTAA1*, and *FveTAR1* exhibited gradual increase in expression from stage 1 ovule to stage 3 ghost (Supplementary Fig. 6a, c), an expression trend opposite that of *FveATHB* genes (Fig. 5b). This opposite expression trend is consistent with the hypothesis that *FveATHBs* may repress the transcription of auxin biosynthesis gene pre-fertilization, and that this repression is gradually lifted from stage 1 to 3 through the action of *FveAGL62/FveAGL80* upon fertilization. To test this hypothesis, Y1H and transient luciferase expression were used to test direct binding of FveATHBs to the *FveYUC10* promoter. In

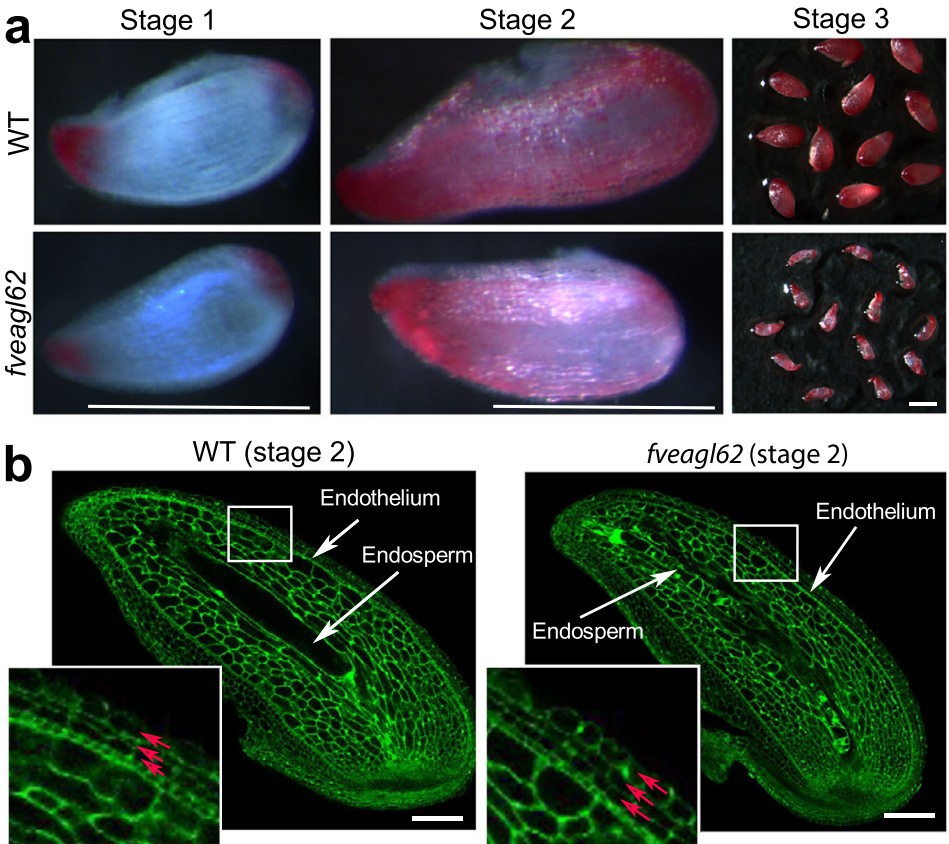

**Fig. 4 Characterization of _F. vesca fveagl62_ mutant seedcoat. a** Vanillin-stained _F. vesca_ WT and _fveagl62_ mutant seeds at the pre-fertilization stage 1 and post-fertilization stages 2–3. The proanthocyanidins in the endothelium layer stains red with the vanillin. **b** Confocal laser scanning microscopy images of WT and _fveagl62_ seeds at stage 2. The enlarged box highlights the three integument cell layers (red arrows) exterior to the endothelium. Scale bars, 500 μm (**a**), 100 μm (**b**).

Y1H, _FveATHB29b_ and _FveATHB30_ each was able to activate _ProFveYUC10::AbAi_ to allow yeast colony formation on media containing AbA$_{500}$ (Fig. 5g). In the tobacco transient expression assay, _FveATHB30_ but not _FveATHB29b_ repressed the expression of _ProFveYUC10::LUC_ (Fig. 5h). Since FveATHB29b and FveATHB30 can heterodimerize (Supplementery Fig. 9) and the ATHB subfamily members were reported to function through homo- and heterodimerization[30], the _FveATHB29b/FveATHB30_ heterodimers, as well as other in vivo heterodimer combinations, may act to directly repress _FveYUC10_ expression.

**Characterization of _FveATHB_ overexpression transgenic lines.** To test the function of _FveATHB29b_ and _FveATHB30_ in strawberry, a reasonable approach is to over-express (OE) _FveATHB_ genes in _F. vesca_, which overcomes functional redundancy among _FveATHB_ family members. Three independent transgenic lines of _FveATHB29b-OE_ and two independent lines of _FveATHB30-OE_, all driven by the UBQ10 promoter, were generated and shown to increase _FveATHB29b_ and _FveATHB30_ expression by 10–20 fold and 3-7 fold, respectively (Supplementary Fig. 10). All _FveATHB29b-OE_ and _FveATHB30-OE_ transgenic lines produced smaller fruits compared to the wild type (Fig. 6a–c). To determine if the small fruit phenotype of _FveATHB-OE_ lines is due to reduced auxin, we applied NAA to the fertilized fruit and observed restoration of fruit size to wild type size (Fig. 6b). To test if the _FveATHB-OE_ shows reduced expression of auxin biosynthesis genes, which may underlie the small fruit size, transcript levels of _FveYUC10_ and _FveTAA1_ in stage 2 seeds of _FveATHB-OE_ lines were quantified by RT-qPCR. _FveYUC10_ and _FveTAA1_

transcript levels were reduced by 2–10 fold in the seeds of _FveATHB-OE_ lines (Fig. 6d, e). Therefore, the small fruits of _FveATHB-OE_ lines are likely caused by a reduction of auxin biosynthesis gene expression.

If the reduced fruit size is due to reduced auxin biosynthesis in _FveATHB-OE_ seeds, some of the seeds should exhibit a phenotype similar to _fveagl62_. Confocal microscopy showed that the majority of _FveATHB-OE_ seeds remained small and contained a tube-like structure filled with dead cells (red arrowheads in Fig. 6f; Supplementary Fig. 11). A small percentage (8.5%) of the endosperms showed precocious cellularization (Supplementary Fig. 11). Sometimes, both cell death and premature endosperm cellularization were present in the same endosperm of _FveATHB-OE_ seeds (see enlarged inserts in Fig. 6f). Since the expression of _FveATHBs_ via the _UBQ10_ promoter occurs in maternal sporophytic as well as gametophytic tissues before and after fertilization, ectopic _FveATHB_ expression is likely causing reduced auxin earlier and in more tissues than what is observed in _fveagl62_, which may lead to endosperm cell death. We showed that application with NAA could partially rescue the endosperm phenotypes in cell death and premature cellularization (Supplementary Fig. 11). Together, our data revealed a previously unknown regulatory module consisting of _AGL62/AGL80_ and _ATHBs_ to regulate auxin biosynthesis in the endosperm of fertilized seeds.

**Discussion**

Auxin from the seeds has been known to induce strawberry fruit development for more than 70 years[3]. Recent studies in diploid

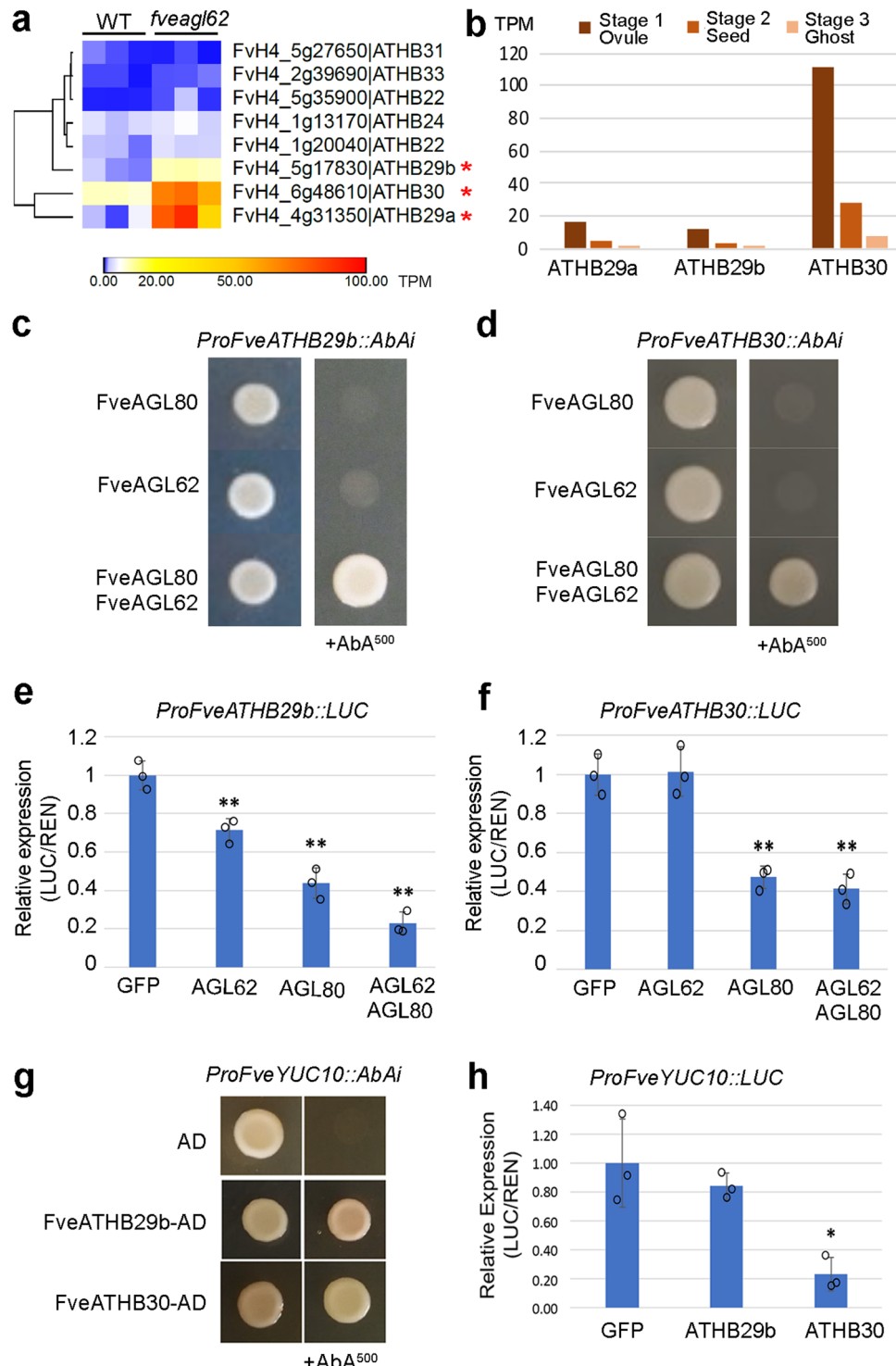

**Fig. 5 *FveAGL62/FveAGL80* regulates a family of *FveATHB* genes in strawberry. a** Hierarchical clustering heatmap showing RNA-seq reads (TPM) in WT and *fveagl62* mutant seeds at stage 2 for a subclass of *FveATHB* genes. Three *FveATHB* genes (marked by *) are more highly expressed in the *fveagl62* mutant seeds than the WT. **b** Bar graphs showing the expression level (TPM) of three *FveATHB* genes at the stage 1 ovules, stage 2 seeds, and stage 3 ghosts in WT based on a prior RNA-seq dataset[5]. **c**, **d** Y1H assays showing binding and activation of promoters of *FveATHB29b* and *FveATHB30* by the combined action of FveAGL62 and FveAGL80. **e**, **f** Transient repression of the LUC reporter driven by the *FveATHB29b* (**e**) or *FveATHB30* (**f**) promoter. Effector proteins are FveAGL62 or FveAGL80 alone or in combination. Significant difference from the negative control (GFP) is marked by ** ($P < 0.01$ by two-tailed Student's *t*-test). Error bars indicate standard deviation. Three biologically independent experiments gave similar results. **g** Y1H assay testing the binding of FveATHBs to *ProFveYUC10::AbAi* in the SD-Leu-Ura medium with or without 500 ng/ml AbA. Empty vector containing the AD serves as a negative control. **h** Transient repression of *ProFveYUC10::LUC* in tobacco leaves. Y-axis indicates the relative expression level of LUC compared to REN. Significant difference from the negative control (GFP) is marked by * ($P < 0.05$ by two-tailed Student's *t*-test). Error bars indicate standard deviation. Three biologically independent experiments gave similar results.

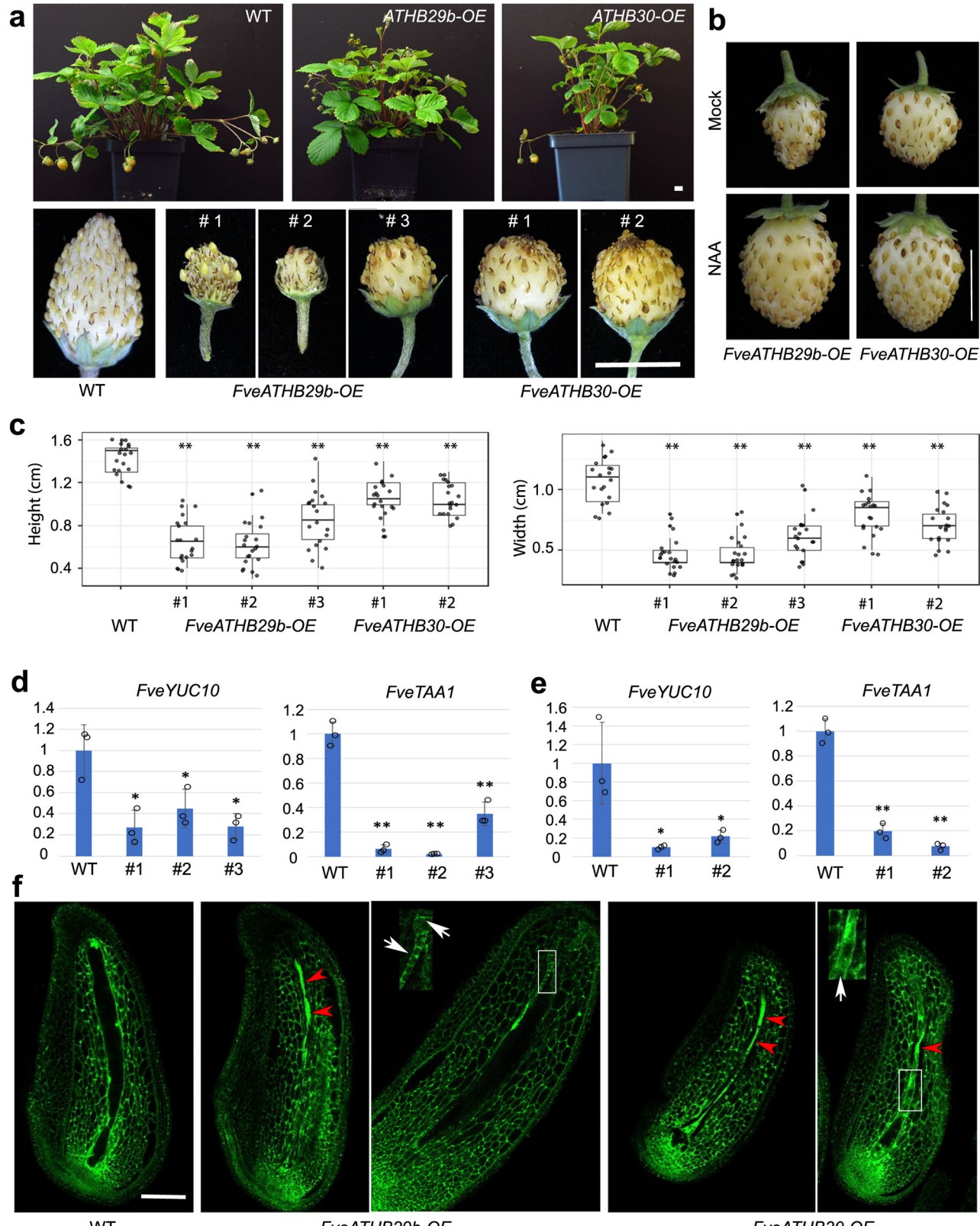

wild strawberry *F. vesca* showed that the fertilization-induced auxin biosynthesis mainly occured in the endosperm[5,12]. However, the molecular mechanism underlying this fertilization-induced auxin biosynthesis in the endosperm is unknown. In this study, we show that the type I MADS-box gene *FveAGL62* is required for auxin biosynthesis in the endosperm and *FveAGL62*

expression is induced by fertilization specifically in the endosperm, thus establishing *FveAGL62* as a key gene that mediates fertilization-induced auxin production for strawberry seed and fruit development.

In addition, we show that *AGL62* plays a conserved function in promoting auxin biosynthesis in strawberry as well as

**Fig. 6 Characterization of *FveATHB-OE* transgenic lines. a** Plant and fruit phenotype of *FveATHB-OE* transgenic lines. Fruits of three *FveATHB29b-OE* lines and two *FveATHB30-OE* lines are shown. **b** Mock- and NAA-treated fruits of *FveATHB29b-OE* and *FveATHB30-OE* lines. **c** Fruit size measurement of WT, three *FveATHB29b-OE* lines, and two Fve*ATHB30-OE* lines. Significant difference from WT is marked by ** (n = 20, P < 0.01 by two-tailed Student's *t*-test). The elements of boxplots and source data are provided in a Source Data file. **d, e** Relative transcript level (Y-axis) of auxin biosynthetic genes *FveYUC10* and *FveTAA1* in stage 2 seeds of WT and *FveATHB29b-OE* (**d**) and *FveATHB30-OE* (**e**) lines. Significant difference (two-tailed Student's *t*-test) from WT is marked with ** (*P* < 0.01) and * (*P* < 0.05). Error bars indicate standard deviation. Three biologically independent experiments gave similar results. **f** Confocal laser scanning microscopy reveals extensive cell death (red arrowheads) and precocious cellularization (white arrows in enlarged inserts) in the endosperm of stage 2 seeds in *FveATHB-OE* lines. Scale bars, 1 cm (**a**, **b**), 100 μm (**f**).

Arabidopsis. Both strawberry *FveAGL62* and Arabidopsis *AtAGL62* exhibit similar expression patterns: their transcripts increase immediately after fertilization in the sexual endosperm and then decline abruptly just before cellularization (Fig. 1a, Supplementary Fig. 1d)[19]. In addition, loss of *AGL62* in strawberry or Arabidopsis causes a similar precocious endosperm cellularization phenotype, and the strawberry *FveAGL62* gene rescues the Arabidopsis *atagl62* mutant phenotypes. Both *fveagl62* and *atagl62* mutants exhibit significantly reduced expression of auxin biosynthesis genes and reduced auxin reporter signal in the endosperm, and application of exogenous auxin partially rescues each mutant's endosperm cellularization defect. These data strongly support AGL62 as a conserved positive regulator of auxin biosynthesis in the sexual endosperm of multiple plant species.

Our finding is supported by previous genetic analysis of Arabidopsis auxin biosynthesis mutants and *atagl62* mutants. Triple loss-of-function mutants in auxin biosynthesis genes had severe endosperm proliferation defects, and exogenous auxin application or central cell-specific activation of auxin biosynthesis genes induced autonomous endosperm development without fertilization[14]. In *fis2* mutants with a defective FIS-PRC2 (Polycomb Repressive Complex 2), auxin is ectopically produced in the central cell, which correlates with central cell replication and autonomous endosperm formation without fertilization[14]. Interestingly, this autonomous endosperm development is suppressed by the *atagl62* mutation[14] and *AtAGL62* was shown to be a direct target of FIS2-PRC2[21]. These genetic analyses suggest that FIS-PRC2 may normally repress auxin and *AtAGL62* to prevent endosperm development before fertilization. Nevertheless, despite their similar role in endosperm development, the relationship between *AtAGL62* and auxin in endosperm development has not been resolved until now.

Based on the findings of this study, we propose a model shown in Fig. 7. Upon fertilization of the central cell and through an as yet undetermined mechanism, the expression of *FveAGL62* and *FveAGL80* is induced in the endosperm. FveAGL62/FveAGL80 heterodimers may directly bind and inhibit the transcription of *FveATHB29b*, *FveATHB30*, as well as other target genes. The decreased level of *FveATHBs* leads to de-repression of *FveYUC10* and other auxin biosynthesis genes, enabling the synthesis and accumulation of active auxin, which is subsequently exported to the receptacle to initiate fruit development.

Other than auxin biosynthesis, *AGL62/AGL80* almost certainly regulates other downstream processes including auxin transport and GA biosynthesis as shown by significantly reduced expression of GA biosynthesis genes (Fig. 2b). In both strawberry and Arabidopsis, *agl62* mutant endosperm still undergoes a few rounds of nuclear division before premature cellularization. These initial nuclear divisions could be stimulated by a low dose of auxin brought in by the sperm cell or by an initial auxin synthesis triggered by the action of fertilization. Therefore, *AGL62* may very well act in a positive-feedback loop to amplify auxin biosynthesis in the endosperm after an initial pulse of auxin that arises independently.

While our work focuses on fertilization-induced auxin synthesis in the endosperm, several prior studies investigated fertilization-induced seedcoat development in Arabidopsis, which revealed that the Arabidopsis seedcoat development relied on the *AtAGL62*-dependent transport of auxin produced from the endosperm[15,20], and that *atagl62* mutants not only failed to develop a seedcoat but also trap and elevate auxin in the endosperm[15]. While we similarly observed a lack of seedcoat development in the Arabidopsis *atagl62* mutants (Supplementary Fig. 7c), we observed a significantly reduced auxin in the *atagl62* mutant endosperm (Fig. 3a, b). We do not know reason behind this difference but note that the studies used different auxin reporters. The prior study used *DR5::VENUS*[15], whereas our study utilizes *R2D2* and *ProAtYUC10::3xnGFP*[27,28].

Another distinction is the apparently normal seedcoat formation in strawberry *fveagl62* mutants, suggesting that strawberry seedcoat development is independent of *FveAGL62* (Fig. 4). We did not find an ABCB transporter among the down-regulated genes in *fveagl62* stage 2 seeds as was observed in *atagl62*. However, an auxin efflux carrier, *FvePIN2* (*FvH4_4g06850*), is down-regulated by tenfold in the strawberry *fveagl62* seeds. As part of the essential role of *FveAGL62* for fruit development, *FvePIN2* could be activated by *FveAGL62* to transport auxin to the receptacle fruit of strawberry. A possible explanation for the different seedcoat phenotype between *atagl62* and *fveagl62* is that the pulses of auxin that initiate seedcoat development could originate from different tissues in strawberry. For instance, the ovary wall of *F. vesca* was found to express auxin biosynthesis genes upon fertilization[5] and could be the tissue that provides the initial auxin for initiating seedcoat development.

In summary, fertilization triggers seed and fruit development through the induction of *AGL62* that leads to auxin synthesis in the endosperm. However, pollination/fertilization-induced auxin synthesis is not exclusive to the endosperm and may occur in other sporophytic tissues as is supported by reports of independent auxin synthesis in the seed integuments[31,32]. Subsequently, newly synthesized auxin not only initiates post-fertilization development in situ but also is transported to other seed and floral tissues to coordinate and sustain the post-fertilization development. Comparative functional analysis of *AGL62* in other plant species will reveal both conserved and species-specific mechanisms for the induction of post-fertilization developmental programs.

## Methods

**Strains and growth conditions**. The *Fragaria vesca* strain Yellow Wonder 5AF7 (YW5AF7)[33] was used as wild type. CRISPR-knockout, overexpression, and GUS reporter studies are all conducted in YW5AF7. Strawberry plants were grown in a growth chamber with a 16-hour light at 25 °C followed by an 8-hour darkness at 22 °C with a relative humidity of 50%.

Arabidopsis WT (Col-0), a T-DNA insertion line in the Col background, SALK_022148 (*atagl62-2*)[19], the *R2D2* reporter (stock number: CS2105637) in the Col background[27] and the *ProAtYUC10::3xnGFP* reporter (stock number: CS69899) in the Col background[28] were all obtained from the Arabidopsis Biological Resource Center (ABRC) (https://abrc.osu.edu/). Seeds of *DR5ver2::GUS* in *F. vesca* (YW5AF7)[12] were obtained from Dr. Chunying Kang.

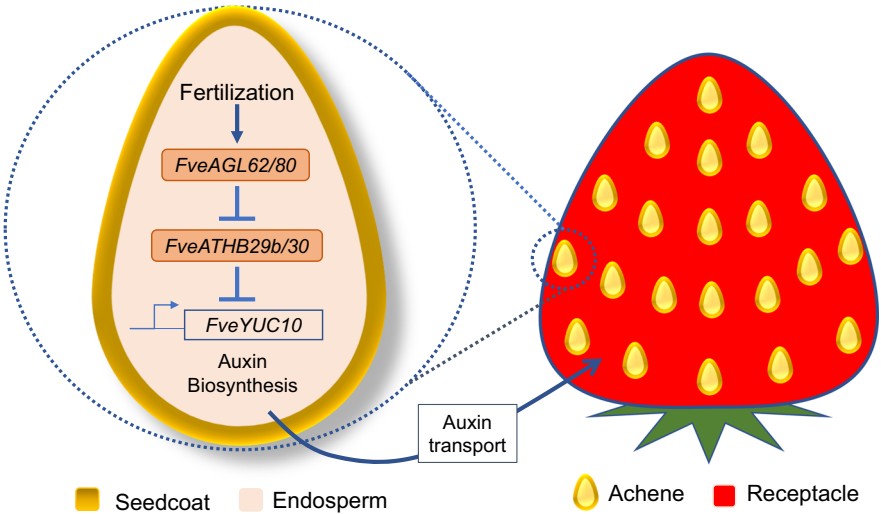

**Fig. 7 A model illustrating the *FveAGL62-FveATHB* module.** Upon fertilization, *FveAGL62* and *FveAGL80* expression is induced in the endosperm and the FveAGL62/FveAGL80 heterodimers act to switch off the transcription of *FveATHB* genes, lifting the transcriptional repression of auxin biosynthesis genes such as *FveYUC10*, leading to the synthesis and accumulation of auxin. Upon being transported to receptacle, auxin stimulates receptacle fruit development.

**Strawberry Gene IDs**. All the genomic and CDS sequences of *F. vesca* genes mentioned in the manuscript can be found in GDR (rosaceae.org) using their gene IDs (genome version 2.0 annotation version 2.0; genome version 4.0 annotation version 2.0) as follows: *FveAGL62* (gene01789; FvH4_2g03030), *FveAGL80* (gene22916; FvH4_6g08460), *FveAGL80L1* (gene18029; FvH4_6g21170), *FveAGL80L2* (gene04949, FvH4_7g09730), *FveAGL80L3* (gene22967; FvH4_6g08570), *FveAGL80L4* (gene23924; FvH4_6g43410), *FveATHB29b* (gene09326; FvH4_5g17830), *FveATHB30* (gene03815; FvH4_6g48610), *FveATHB22* (gene01920; FvH4_1g20040), *FveYUC5* (gene32686; FvH4_2g14550), *FveYUC10* (gene27796; FvH4_2g24750), *FveYUC11* (gene06886; FvH4_4g17980), *FveTAR1* (gene31791/gene37056; FvH4_5g05900), *FveTAA1* (gene03586; FvH4_4g25850), *FveGA20OX1c* (gene19436; FvH4_7g12610), *FveGA20OX1d* (gene13360; FvH4_7g28670), *FveGA3OX1b* (gene01060; FvH4_2g30010), *Fve-GA3OX1c* (gene01059; FvH4_2g30020), *FvePIN2* (gene12312; FvH4_4g06850), *FvePP2a* (gene03773; FvH4_4g27700).

**CRISPR and GUS constructs and transgenic strawberry generation**. To generate the *F. vesca agl62* mutants, gRNA1 (GGGGTGCGCAAGGCGAGCCAGGGG) or gRNA3 (CTATCACAGACTCCACGCAGGGG) targeting the coding region of *FveAGL62* were inserted into the JH4 entry vector and then incorporated into the binary vector JH19 via gateway cloning[34]. To generate the *fveagl80* mutant, one gRNA (CGATGGACTCCCAACGGGGAAGG) targeting the coding region of *FveAGL80* was inserted into the JH4 entry vector and then incorporated into the binary vector JH19. To confirm editing, genomic sequences spanning the target site of respective genes were amplified by PCR and sequenced. All primers used in this study are listed in Supplementary Table 1.

For *ProFveAGL62::FveAGL62-GUS* and *ProFveAGL80::FveAGL80-GUS*, *ProFveAGL62::FveAGL62* (1512-bp upstream and the 660-bp coding region of *FveAGL62*) and *ProFveAGL80::FveAGL80* (2461-bp upstream and 834-bp coding region of *FveAGL80*) were PCR amplified and cloned into pMDC162 binary vector[35], in which the GUS reporter is fused at the C-terminus of the gene. The constructs were transformed into agrobacterium strain GV3101.

For *F. vesca* transformation, calli are induced from cut cotyledons of *F. vesca* (YW5AF7) on the 5++ medium (1 x MS, 2% sucrose, 3.4 mg/L benzyl adenine, 0.3 mg/L indole-3-butyric acid (IBA), 0.7% phytoagar, pH 5.8) for two-to-three weeks in the dark. The calli are then incubated with the agrobacterium GV3101 harboring the construct in the dark for 1 h in the co-cultivation buffer (1 x MS, 2% sucrose, 0.4 mg/ml acetosyringone) with gentle shaking. The infected calli were kept on the 5++ medium for three days in the dark, then washed with sterile water and put onto the MS medium with antibiotics (250 mg/L timentin + 250 mg/L carbenicillin), and grown in the dark for 2 weeks. Transformed calli were selected on the 5++ medium with 250 mg/L timentin, 250 mg/L carbenicillin and 4 mg/L hygromycin. The calli were moved to fresh medium (5++ medium with the same three antibiotics) every 2-to-3 week until shoots appear. When the shoots grow larger, they are transferred to the rooting medium (0.5 x MS, 0.01 mg/L IBA, 2% glucose, 0.7% phytoagar, pH 5.8) with 4 mg/L hygromycin. After about 1–2 months, the plants with roots are transferred to soil and genotyped[5,36]. Nine and 12 independent transgenic lines of *ProFveAGL62::FveAGL62-GUS* and *ProFveAGL80::FveAGL80-GUS*, respectively, were obtained and analyzed.

For overexpression, full-length CDS of *FveATHB29b* and *FveATHB30* were PCR amplified from cDNA and assembled into vector JH23 at the digestion sites of KpnI and PacI through Gibson cloning[37,38]. Thus, *FveATHB29b* and *FveATHB30*

were driven by the Arabidopsis Ubiquitin 10 promoter. After transformation into *F. vesca*, 10 and 12 independent lines of *FveATHB29b-OE* and *FveATHB30-OE* respectively were confirmed by PCR genotyping.

**Analyses of Arabidopsis *atagl62* mutants**. The full-length *FveAGL62* ORF (663 bp) was PCR amplified and assembled with the upstream and downstream fragments of *AtAGL62*. The 2074 bp upstream fragment of *AtAGL62* contains the promoter and 5'UTR, while the 1054 bp downstream fragment contains the 3'UTR of *AtAGL62*. This *ProAtAGL62::FveAGL62:terAtAGL62* chimeric gene was cloned into pMDC99[35] and transformed via floral dip into the Arabidopsis *atagl62* (−/+) heterozygous plant. After germination on ½ MS medium with 30 μg/ml hygromycin, the hygromycin-resistant seedlings were further genotyped using Salk022148/*atagl62-2* T-DNA genotyping primers (Supplementary Table 1). 12 independent *agl62-2*−/− homozygote plants containing the *ProA-tAGL62::FveAGL62:terAtAGL62* transgene were obtained and analyzed.

To analyze the auxin reporters (*R2D2* or *ProAtYUC10::3xnGFP*) in *atagl62-2* seeds, *atagl62-2* (−/+) plants were crossed with plants harboring *R2D2* or *ProAtYUC10::3xnGFP*. The F1 plants containing both the *atagl62-2* mutation and the reporter gene were confirmed by PCR and GFP fluorescence. F2 plants homozygous for the reporter gene and heterozygous for the *atagl62-2* mutation were obtained by genotyping and segregation. Homozygous *ProAtYUC10::3xnGFP* is necessary to avoid impacts of parental imprinting. Fertilized seeds from F2 plants were analyzed under a Leica SP5X Confocal microscope (Leica Co. USA).

**Seed and fruit characterization**. To dissect and image seeds, ovules/seeds at stages 1 (opening flower), 2 (2–4 DPA), and 3 (6–7 DPA) were dissected out of achenes using tweezers and then imaged with a Zeiss Stemi SV 6 microscope equipped with an AxioCam digital camera.

For hormone treatment of strawberry, *fveagl62* receptacles were treated with 500 μM NAA, 500 μM GA, or a combination of NAA and GA. A drop of Tween-20 was added to 10 ml solution. 10 ml water with a drop of Tween-20 was used as the mock control. These treatments were repeated every two days until day 10 when the images were taken. *FveATHB-OE* receptacles were treated with 500 μM NAA or mock control. These treatments were repeated every two days until day 30 when the images were taken.

To treat Arabidopsis seeds with 2,4-D, *atagl62-2* (−/+) siliques at 1DPA were submerged in 200 μM 2,4-D solution for 30 sec[15]. For mock treatment, 0.05% Tween-20 solution was used instead. The seeds were collected the next day and processed for confocal imaging.

For confocal imaging, stage 2 seeds of *F. vesca* were dissected out from the achenes prior to fixation. The 2 DPA Arabidopsis seeds were dissected out of the siliques Seeds were fixed in 4% glutaraldehyde (in 12.5 mM cacodylate, pH 6.9), and kept under a vacuum for 2 h. After fixation, the seeds were dehydrated through an ethanol series [30, 50, 70, 90, 100% (v/v)] at 15 min per step. The dehydrated seeds were clarified in 2: 1 (v/v) benzyl alcohol for at least 1 h before imaging[39]. A Leica SP5X Confocal microscope (Leica Co. USA) with 488 nm excitation was used to image autofluorescence of the seeds. For observing fluorescent reporters in Arabidopsis seeds, eGFP of *ProAtYUC10::3xnGFP* was excited at 488 nm and detected at 498-530 nm. For observing the R2D2 fluorescent reporter in Arabidopsis seeds, a Leica STELLARIS DLS Confocal microscope (Leica Co. USA) was used. DII-VENUS was excited at 515 nm and detected at

520–559 nm and mDII-Tdtomato was excited at 554 nm and detected at 562–726 nm. The pixels of VENUS and Tdtomato fluorescence in endosperm nuclei were measured by the LAS X software (Leica Co. USA) and used to calculate the DII/mDII ratio.

For seedcoat vanillin staining, seeds of Arabidopsis or strawberry were dissected and mounted in vanillin staining solution, 1% (w/v) vanillin (4-hydroxy-3-methoxybenzaldehyde) in 6 N HCl, for 30 min and then imaged with a Zeiss Stemi SV 6 microscope equipped with an AxioCam digital camera.

**RNA-seq analysis**. Strawberry stage 2 seeds were dissected out of achenes. About 50 seeds were collected for each biological replicate and three replicates were prepared for each genotype. Total RNA was extracted using the Monarch Total RNA Miniprep Kit (NEB) following the manufacturer's instructions. A total of 0.5 to 2 μg RNA per sample was sent to the Novogene Corporation Inc for library preparation and sequencing with Illumina Platform. About 43 to 60 million PE150 raw reads per sample were obtained.

For read quantification, Salmon (v0.11.2)[40] was applied to estimate the transcript abundance. Salmon index was generated using Fragaria_vesca_v4.0.a2.transcripts.fa as the reference transcriptome with parameters -k 31 and --keepDuplicates. The reference transcriptome was retrieved from Genome Database for Rosaceae (GDR, https://www.rosaceae.org/species/fragaria_vesca/genome_v4.0.a2)[26,41]. The paired-end RNA-Seq reads were quantified in Salmon mapping-based mode. The --seqBias flag was used to turn on sequence-specific bias correction. The isoform expression was summarized into gene level by Tximport (v1.10.1)[42].

For differential expression analyses, DESeq2 (v1.22.2)[43] was utilized to identify differentially expressed genes ($padj < 0.05$ and $|\log_2 FoldChange| > 1$) between WT and *fveagl62*. The longest strawberry protein variants from *Fragaria vesca* annotation v4.0.a2 were searched against Arabidopsis protein database via OmicsBox (v1.2.4) built-in BLAST[44]. The BLAST database was constructed using the longest protein sequences derived from TAIR10_pep_20101214 (https://www.arabidopsis.org/download/index-auto.jsp?dir=%2Fdownload_files%2FProteins%2FTAIR10_protein_lists) with E-value cutoff set to 0.001.

For Gene Ontology (GO) enrichment analysis, GO terms were assigned to the longest strawberry proteins (v4.0.a2) by OmicsBox (v1.2.4) based on BLAST and InterProScan results[45,46]. The protein sequences were searched against Swiss-Prot database (uniport_sprot.fasta, https://www.uniprot.org/downloads)[47] locally using OmicsBox built-in BLAST. InterProScan was executed within OmicsBox to identify the protein domains with default settings. TopGO (v2.34.0)[48] was employed to conduct GO enrichment analysis. The GO terms with $P$ value ≤ 0.05 in Fisher's exact test were considered to be significant.

For gene expression visualization, the heatmaps were made using the website Morpheus (https://software.broadinstitute.org/morpheus/) with the Hierarchical Clustering option.

**Phylogenetic tree building**. Full Pfam alignment files PF00319 and PF01486 (https://pfam.xfam.org/) were downloaded and analyzed by HMMER (version 3.3) (http://hmmer.org/)[49] to identify the strawberry and Arabidopsis proteins with the MADS-box domain or K-box domain. The proteins that only have a MADS-box domain were considered as type I MADS-box proteins. Protein sequences of strawberry type I MADS-box genes were downloaded from GDR (www.rosaceae.org/) and those of Arabidopsis genes were downloaded from TAIR (https://www.arabidopsis.org/). The sequence alignment was made using Clustal Omega (http://www.ebi.ac.uk/Tools/msa/clustalo)[50]. The unrooted phylogenetic tree was constructed using the online tool PhyML 3.0 (http://www.atgc-montpellier.fr/phyml/) with the neighbor-joining statistical method.

**RNA extraction, cDNA synthesis, and RT-qPCR**. Total RNA was extracted from stage 1 (unfertilized ovules), stage 2 (fertilized seeds), and stage 3 (fertilized ghost) following the Cetyl Trimethyl Ammonium Bromide (CTAB) RNA extraction method[51] and reverse transcribed to cDNA using SuperScript IV VILO Master Mix (Invitrogen, USA). RT-qPCR was performed using BioRad CFX96 Real-time system and SYBR Green PCR MasterMix (Applied Biosystems). The relative expression level was analyzed using a modified $2^{-\Delta\Delta CT}$ method[52]. For all RT-qPCRs, *FvePP2a* (FvH4_4g27700) was used as the internal control[36]. The RT-qPCR experiments reported in Figs. 2, 5, 6 and Supplementary Figs. 6, 11 are from one representative biological experiment (with three technical repeats). Biological replicates (2 to 3 times) gave similar results.

**Testing protein-protein and protein-promoter interactions**. For BiFC assays, full-length CDS of genes of interests were PCR amplified and ligated into either PXY105 (cYFP) at BamHI and XbaI sites or PXY106 (nYFP)[53] at BamHI and XbaI sites through Gibson cloning[37]. BiFC constructs were transformed into Agrobacterium GV3101. Agrobacteria were harvested when OD600 reached 1.5, and resuspended in a solution (10 mM MgCl₂, 10 mM MES (2-(N morpholine) ethanesulfonic acid) pH 5.6, and 200 μM acetosyringone) to a final OD600 of 0.8[54]. The agrobacterial mixture was infiltrated into young *Nicotiana benthamiana* leaves. After 48 h, the YFP fluorescence signal was visualized and photographed using a Leica SP5X confocal microscope (Leica Co. USA).

For Y2H assays, full-length CDS of genes of interest were PCR amplified and ligated into pGADT7 (Clontech Inc.) at the EcoRI/BamHI digestion sites or pGBKT7 (Clontech Inc.) at the EcoRI/BamHI digestion sites by Gibson assembly[37]. Yeast strain PJ69-4A was transformed using the LiAc/PEG method[55].

For Y1H assays, the Y1H constructs and procedures were performed based on the Matchmaker Gold Yeast One-Hybrid Library Screening System User Manual (PT4087-1). Full-length CDS of genes of interest were PCR amplified and ligated into pGADT7 (Clontech Inc.) at the EcoRI/BamHI digestion sites by Gibson assembly[37]. The target promoter sequence was amplified and ligated into pAbAi (Clontech Inc.) at the KpnI/SalI digestion sites by Gibson assembly[37]. Yeast strain Y1HGold was transformed using the LiAc/PEG method[55].

For transient luciferase assays, the promoters of *FveYUC10*, *FveATHB29b*, and *FveATHB30* were PCR amplified from genomic DNA, first Gibson assembled into a modified gateway entry vector LEI2[38], and then into the destination vector pLAH-LARm vector[56] by LR recombination. The full-length CDS of *FveAGL62* and *FveAGL80* were PCR amplified from genomic DNA and assembled into the vector JH23 at the digestion sites of KpnI and PacI through Gibson cloning[37,38]. The constructs were transformed into agrobacterium GV3101. The agrobacterial solution was then infiltrated into young *Nicotiana benthamiana* leaves. At 48 h after agroinfiltration, the tobacco leaf tissue was collected and treated using the Dual-Luciferase Reporter Assay System (Promega). The luminescence activities of the firefly luciferase (LUC) and the Renilla luciferase (REN) were measured with a TD-20/20 Luminometer (Turner Designs). The Renilla luciferase gene (35 S:REN) was used as a control to normalize the LUC readout. The assay was performed as previously described[57]. Each experiment was repeated two to three times with one representative result shown in Fig. 5e, f, h. All primers for the above assays are listed in Supplementary Table 1.

**Reporting summary**. Further information on research design is available in the Nature Research Reporting Summary linked to this article.

## Data availability

The RNA-seq data have been submitted to SRA with the accession number PRJNA774354. Source data are provided with this paper.

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

## Acknowledgements

We thank Dr. Chunying Kang (Huazhong Agricultural University) for *DR5ver2::GUS* seeds of *F. vesca*. We also thank Amy Beavan and the Dept. of Cell Biology and Molecular Genetics Imaging Core at the University of Maryland for the Leica SP5X Confocal microscope. We thank Dr. Heven Sze for helpful comments of the manuscript. This work has been supported by the NSF award IOS-1444987 to Z.L., NSF award DEG-1632976 to M.L., Maryland Summer Scholars Award to M. P., and the University of Maryland CMNS Dean's Matching Award-NIH T32 (Molecular and Cell Biology) to D.J.

## Author contributions

Z.L. and L.G. designed the project. L.G. performed most of the experiments. X.L. performed strawberry transformation. M.L. analyzed the RNA-seq data. M.P. contributed to transgenic plant analysis. L.G., D.J. and Z.L. analyzed and interpreted the data. L.G. and Z.L. prepared the figures and wrote the manuscript.

## Competing interests

The authors declare no competing interests.
