## [Peer Review File · Nature Communications]

Mechanism of fertilization-induced auxin synthesis in the endosperm for seed and fruit developmentREVIEWER COMMENTS

Reviewer #1 (Remarks to the Author):

In this manuscript by Guo et al, the authors aim to understand the molecular mechanisms driving post-fertilization auxin biosynthesis in the endosperms of wild strawberry and Arabidopsis. The production of auxin in fertilized seeds had previously been shown to drive the differentiation of the endosperm of the surrounding maternal tissues, but the mechanisms that activate the expression of auxin biosynthesis genes remained unclear. Here, the authors demonstrate that the transcription factor AGL62 regulates the expression of auxin biosynthesis genes and that *agl62* mutations lead to seed arrest, including precocious cellularization of the endosperm. This phenotype can be partially rescued by the exogenous application of auxin in strawberry seeds, but not so clearly in Arabidopsis. They also demonstrate that the FveAGL62 gene can complement the developmental defects of the Arabidopsis *agl62* mutant. Finally, the authors propose that ATHB transcription factors act downstream of AGL62 to regulate the expression of auxin biosynthesis genes. Overall this is an interesting and clear manuscript. Most of the data seems solid and is well presented. However, I have some reservations on the claims by the authors that exogenous auxin rescues the endosperm cellularization phenotype of the Arabidopsis *agl62* mutant and, more importantly, on the claim that ATHBs regulate the expression of auxin biosynthesis genes during seed development. I detail these points, and others, below.

Major points:

- 1 - Line 145/Figure 1E – The authors conclude that there are no parent-of-origin effects for the *fveagl62* mutation. However, fruits resulting from both reciprocal crosses seem smaller than the selfed WT and some achenes seem aborted/brownish. Could the authors clarify this? Furthermore, if the parent-of-origin effect cannot be fully assessed using these crosses, I would suggest doing reciprocal crosses of the AGL62:GUS reporter with WT, to test if FveAGL62 is expressed from both parents.
- 2 - Figure 3 – Regarding the assays with the DII reporter, it is not clear to me if the authors used the R2D2 sensor published by Liao et al, 2015, in Nature Methods (they cite this paper, but I do not understand if this is the actual reporter that they used). If so, this sensor carries two fluorescent fusion proteins, DII:VENUS and mDII:Tdtomato. The Tdtomato protein is used as a control, and auxin activity is measured as a VENUS/Tdtomato ratio. However, the authors do not show the Tdtomato data. This is particularly important because the DII data conflicts with the report by Figueiredo et al, 2016, in Elife, where *agl62* mutant seeds have higher auxin activity than WT ones. The differences between the two studies could be due to the use of different reporters, DR5 and DII. But this should be cleared up, or at least discussed in the manuscript.
- 3 - Page 13 – first paragraph – The authors claim that the precocious endosperm cellularization characteristic of the Arabidopsis *agl62* mutant can be rescued by exogenous application of auxin. I do not think there is enough evidence for this, especially given the small sample sizes: around 50 seeds. In particular and given that the authors had to work with heterozygous plants, because *agl62* is zygotic lethal, only 25% of seeds per silique are homozygous *agl62*^{-/-}. Hence, within the same silique, how did the authors distinguish between a WT seed (genotypically AGL62/AGL62 or AGL62/*agl62*) and an *agl62*/*agl62* seed with a rescued endosperm? These seeds should look the same.
- 4 - Line 329 – I would be cautious with the conclusion that FveAGL62/80 are not direct regulators of FveYUC10. The two transcription factors likely act as heterodimers. Therefore, an untargeted Y1H screen would likely not be sufficient to pick them up. Only actively expressing both AGLs in a yeast strain containing the FveYUC10 promoter fragment would be sufficient to draw conclusions. I do not contest the data that FveYUC10 is regulated by ATHBs, but would not rule out that AGL62/80 could also regulate this gene directly.
- 5 - Fig. 6 - It is clear that the overexpression of ATHBs leads to reduced fruit size in strawberry. However, no data is shown on whether seed development is also affected in these lines. If the authors want to make the point that ATHBs are regulators of auxin biosynthesis genes in the endosperm, then the overexpressor lines should produce seeds with the same defects as the *fveagl62* mutants. There is no indication that this is the case. There is also no evidence that ATHBs are expressed in the

endosperm.

6 - Page 16 – the claim that the ATHBs function downstream of AGL62/80 to regulate auxin biosynthesis is not fully tested. In particular, the authors do not show whether auxin activity as measured by DR5 is affected in the seeds of ATHB over-expression lines. Nor whether application of auxin can rescue the ATHB overexpressor phenotype, like it does to the fveagl62 mutant. Furthermore, there is no evidence for a direct link between AGL62, ATHB and auxin production. I understand that this would require knocking-out ATHBs in the fveagl62 mutant background, and that this is no easy task in strawberry. However, I do not think the authors have enough data to back up their claim of a regulatory module consisting of AGL62/80, ATHBs and auxin biosynthesis.

Other points:

- 1 - Figure 1 – because the structure of the strawberry seed is not as known as the one from Arabidopsis, it would be helpful to have a legend indicating where the seed tissues are located. The authors mention that the GUS reporters are specifically expressed in the endosperm, but it is not clear to me where the endosperm starts and ends.
- 2 - The authors refer to strawberry seeds at stage 1 and 2, meaning pre- and post-fertilization. However, at stage 1 they should be referred to as "ovules" rather than "seeds".
- 3 - Page 10, first paragraph – DR5 expressed is used as a measure of auxin activity, not the actual amount of auxin. Therefore, "auxin level" should be corrected to "auxin activity".
- 4 - Line 226/227 – how many seeds were assayed in the mock-treated fveagl62 assay?
- 5 - For the transactivation assays of Fig 5 and 6, what was used as an internal control, to normalize the LUC readings?
- 6 - Figure S7 – The negative controls are missing for ATHB30.
- 7 - Which promoter was used for the overexpression of ATHBs in strawberry? This is not mentioned in the manuscript and I could not find information on the vector JH23 online.

Reviewer #2 (Remarks to the Author):

The present manuscript by Guo et al on the " Mechanism of fertilization-induced auxin synthesis in the endosperm for seed and fruit development" attempts to demonstrate the mechanisms of how fertilization induces auxin biosynthesis in the endosperm to promote fleshy fruit initiation in strawberry. In the previous paper, Kang et al have reported that AtAGL62 plays an important role in the regulation of auxin synthesis in the Arabidopsis. Throughout the manuscript, several strawberry FveATHB genes were identified as downstream targets of FveAGL62 and act to repress auxin biosynthesis. This research topic itself is highly valuable for general readers of "Nature communication". However, it seems to me that there are some critical flaws with the manuscript. Biochemical and genetic evidence should be necessary to support the authors' claims. The following are some of the critical comments and experiments.

1. Line111-112: Three mutant strains of FveAGL62 were obtained, but only one was selected in the follow-up experiment. I wonder why the other two mutants don't do follow-up experiments? If the phenotypes of the three mutant lines are consistent, wouldn't it be more illustrative?
2. Line138-140: Authors carried out Y2H and BiFC assays which showed that FveAGL62 and FveAGL80 interacted with each other. I think that author should provide more experimental evidence, for example, Co-IP or Pull-down.
3. Line141-143: These data suggest that FveAGL62 and FveAGL80 proteins may act together as heterodimers similar to their Arabidopsis homologs. They constructed fveagl62-1/-4, fveagl80-1/-1 double mutants, which exhibited a similar phenotype as the fveagl62-1/-4 single mutant. But, the fveagl80 -1/-1 mutant exhibited normal vegetative growth and developed fruit similarly to wild type (Figure S2C). I wonder whether the heterodimer of two proteins leads to the emergence of double mutant phenotype, or whether fveagl62-1/-4 gene knockout is the main factor. I think that experimental evidence supporting this conclusion is too weak.
4. Line172-176: The author identified six genes related to auxin synthesis by RNA-seq data, but the

author only selected FveTAR1 and FveYUC10 for qRT-PCR verification. Why did the author only select two genes FveTAR1 and FveYUC10.

5. Line181-185: They found that the transcripts of many auxin and GA biosynthesis genes exhibited a gradual increase from stage 1 to stage 3 in the seeds, consistent with their roles in fertilization-induced auxin biosynthesis in seeds. I think that authors should verify the expression of these genes in seeds by qRT-PCR.

6. Line329-331: 1) The Y1H experiment described in the results confirms that FveAGL62 or FveAGL80 cannot bind FveYUC10 directly, suggesting an indirect effect of FveAGL62 on auxin biosynthesis. However, there is no corresponding figures. The author should put the result figures of Y1H experiment in the manuscript or supplement data. 2) In addition to FveYUC10 gene, the author should use Y1H experiment to verify the relationship between FveAGL62 and other auxin synthesis genes identified in this paper.

7. Line 333-335: The authors introduction that one member of this subfamily was previously found to bind the FveYUC10 promoter in a Y1H screen. This experimental result is found in this manuscript or reported in previous studies. If it is previous research, references should be added.

8. Line 341-346: The author only uses LUC assay to prove that FveAGL62 / FveAGL80 directly represses the expression of these FveATHBs, and there is too little experimental evidence. The authors should supplement EMSA or Y1H experiments to prove that FveAGL62/FveAGL80 can bind the promoter of FveATHBs and repress their expression.

9. Line 387-391: In this paper, the author obtained the overexpression line of FveATHBs. It was proved that AGL62/AGL80 and ATHB to regulate auxin biosynthesis in the endosperm of fertilized seeds by measuring the size of transgenic fruit and the expression level of auxin synthesis related genes. I suggest that the phenotype of endosperm cellularization should be supplemented in FveATHB-OE.

10. In general, the author only used knockout lines to study the function of AGL2 gene, while FveATHBs gene only used overexpression lines. If the article supplemented overexpression and knockout lines, the experimental data would be more sufficient.

Reviewer #3 (Remarks to the Author):

The manuscript entitled "Mechanism of fertilization-induced auxin synthesis in the endosperm for seed and fruit development" reports a novel regulatory pathway for auxin induction in the endosperm after fertilization. In this work, authors show us some interesting findings regarding to auxin biosynthesis in strawberry and Arabidopsis endosperm. (1) Auxin synthesis is induced in the endosperm after fertilization. (2) MADS box gene AGL62 plays a critical role in the activation of auxin synthesis via AGL62-ATHB pathway.

Although the role of AGL62 in suppressing endosperm cellularization has been reported in Arabidopsis, its role in regulating endosperm development and the conservation of this mechanism among different species, as well as the downstream pathways of AGL62 remain largely unknown. This work provides novel knowledge of a fertilization triggered mechanism for activation of auxin synthesis in the endosperm.

Together with previous works about Arabidopsis AGL62, this newly added information will enhance our understanding about endosperm development in angiosperms, especially about molecular linkage between fertilization and endosperm initiation. The researchers in the field of plant development and horticulture will be interested in this work. Therefore, it is worthy to be known by readers.

Here, I list several concerns that may help authors to improve the manuscript before publication:

1. In Arabidopsis, previous work demonstrated that auxin response increases in the ovules after fertilization due to up-regulated auxin biosynthesis in the integuments (Robert et al., Nature Plants, 2018). Here, authors revealed that fertilization induces auxin biosynthesis in the endosperm. Some

additional discussion in the manuscript may help readers to understand sequential of the auxin biosynthesis activation in different seed tissues in relation to fertilization.

2. In the manuscript, authors state that fertilization of the central cell leads to increased auxin biosynthesis. I am not sure whether fertilization in the central cell is a required step for auxin biosynthesis. It is very interesting to know if fertilization is indeed a required step for auxin biosynthesis activation in the endosperm. How about auxin biosynthesis activation in the autonomously developed endosperm as that in PRC2 mutants? What is the real trigger for auxin biosynthesis activation?

3. Authors demonstrates that auxin is responsible for the phenotype of endosperm in *agl62* mutants. Are there any developmental defects in the endosperms of mutants with auxin biosynthesis defects? Related evidences will further confirm the conclusion in the present manuscript. In addition, GA biosynthesis related genes were also downregulated in *agl62* mutants. Does GA also play a critical role in endosperm development?

4. In Figure S2, authors show us that FveAGL62 could interact with FveAGL80 in yeast. Besides FveAGL80, there are another four FveAGL80-Like genes in the strawberry genome. Could FveAGL62 also interacts with other FveAGL80-Like genes?

5. The relationship between AGL62 and ATHB, ATHB and auxin biosynthetic genes were investigated by yeast system or transient tobacco system. In vivo Chip-PCR will further confirm the relationship between AGL62 and its downstream genes.

6. In this work, the whole seeds, including seed coats, endosperm and embryo, but not endosperm only, were collected for RNA-seq and RT-qPCR. Seed coats and embryos may influence the analysis and interpretation of the results. Is it possible that auxin is synthesized in the seed coats and transports to the endosperm?

7. Authors show us an important finding that Strawberry FveAGL62 could rescue Arabidopsis *atagl62* mutant seeds. More detailed evidences such as how many transgenic lines show normal seed setting as wild type plants and detailed endosperm phenotype analysis in the FveAGL62 transgenic lines will underpin this new finding. In addition, comparative analysis of AGL62 genes in different representative plants during evolution will provide useful information about AGL62.

8. In the method section, the detailed information about FveATHB29b and FveATHB30 overexpression is missing. Which promoter was used to drive FveATHB29b and FveATHB30 overexpression in the endosperm? Is it exclusively active in the endosperm?

9. In Figure 6, only the phenotype of fruits was analyzed in FveATHB29b-OE and FveATHB30-OE transgenic lines. Detailed phenotype analysis of the endosperm in FveATHB29b and FveATHB30 overexpression plants will help to confirm the AGL62- ATHB29 pathway.

10. In Figure 6, authors demonstrated that auxin biosynthesis genes such as YUC10 and TAA1 were significantly down-regulated in FveATHB29b-OE and FveATHB30-OE transgenic plants. Since treatment of *fveagl62* mutants with GA could also lead to fruit and seed enlargement, whether GA biosynthesis pathways were also affected in FveATHB29b and FveATHB30 overexpression lines?

11. In the tobacco transient assay, FveATHB30, but not FveATHB29b, could repress the expression of ProFveYUC10::LUC. But the expression levels of YUC were significantly reduced in both FveATHB29b and FveATHB30 overexpression lines. These two results seem not in line with each other.

12. When comparing the fruit phenotype in FveATHB29b-OE and FveATHB30-OE transgenic lines, fruit development seems affected more seriously in FveATHB29b-OE. But the ability of ATHB30 in

repressing the expression of downstream auxin biosynthetic genes, such as YUC10 , is stronger than ATHB29b as shown in Figure 6b,e,f. Potential reasons for the differences could be discussed in the manuscript.

Dear Reviewers,

Thank you very much for your helpful comments and suggestions. This revised manuscript has incorporated new experimental data into Figure 1h, Figure 2c, Figure 3a, c, d, Figure 5c, and Figure 6b, f, added four new supplementary figures (Figure S3, Figure S4, Figure S9, Figure S12), and revised three supplementary figures (Figure S6, Figure S8, Figure S10). These new data and revisions in response to the many helpful suggestions by you have strengthened and improved the manuscript significantly.

Below is a point-by-point response. Corresponding changes in the manuscript text and figure legends are highlighted in red.

Thank you so much for your time and your consideration.

Sincerely,

Zhongchi Liu

Response to reviewer comments

Reviewer #1 (Remarks to the Author):

In this manuscript by Guo et al, the authors aim to understand the molecular mechanisms driving post-fertilization auxin biosynthesis in the endosperms of wild strawberry and Arabidopsis. The production of auxin in fertilized seeds had previously been shown to drive the differentiation of the endosperm of the surrounding maternal tissues, but the mechanisms that activate the expression of auxin biosynthesis genes remained unclear. Here, the authors demonstrate that the transcription factor AGL62 regulates the expression of auxin biosynthesis genes and that *agl62* mutations lead to seed arrest, including precocious cellularization of the endosperm. This phenotype can be partially rescued by the exogenous application of auxin in strawberry seeds, but not so clearly in Arabidopsis. They also demonstrate that the *FveAGL62* gene can complement the developmental defects of the Arabidopsis *agl62* mutant. Finally, the authors propose that ATHB transcription factors act downstream of AGL62 to regulate the expression of auxin biosynthesis genes.

Overall, this is an interesting and clear manuscript. Most of the data seems solid and is well presented. However, I have some reservations on the claims by the authors that exogenous auxin rescues the endosperm cellularization phenotype of the Arabidopsis *agl62* mutant and, more importantly, on the claim that ATHBs regulate the expression of auxin biosynthesis genes during seed development. I detail these points, and others, below.

Thank you for the encouraging comments. Both of your concerns (underlined) are addressed in the responses below (1.3 and 1.5).

Major points:

1 - Line 145/Figure 1E – The authors conclude that there are no parent-of-origin effects for the *fveagl62* mutation. However, fruits resulting from both reciprocal crosses seem smaller than the selfed WT and some achenes seem aborted/brownish. Could the authors clarify this? Furthermore, if the parent-of-origin effect cannot be fully assessed using these crosses, I would suggest doing reciprocal crosses of the AGL62:GUS reporter with WT, to test if FveAGL62 is expressed from both parents.

1.1 Small variations in fruit size and lack of development or aborted development in some achenes occur often in our growth environment even for wild type plants. It is due to variations in environment (pests, humidity, etc.).

*1.2 Based on your suggestion to fully assess the parent-of-origin, we performed reciprocal crosses of the pFveAGL62:GUS transgenic plant with WT nontransgenic plant and examined GUS expression in F1 seeds. The data is added to **Figure 1h**; the GUS reporter is expressed in the F1 endosperm irrespective of which parent provides the pFveAGL62:GUS transgene. Therefore, there is no parent-of-origin effect for FveAGL62 expression.*

2 - Figure 3 – Regarding the assays with the DII reporter, it is not clear to me if the authors used the R2D2 sensor published by Liao et al, 2015, in Nature Methods (they cite this paper, but I do not understand if this is the actual reporter that they used). If so, this sensor carries two fluorescent fusion proteins, DII:VENUS and mDII:Tdtomato. The Tdtomato protein is used as a control, and auxin activity is measured as a VENUS/Tdtomato ratio. However, the authors do not show the Tdtomato data. This is particularly important because the DII data conflicts with the report by Figueiredo et al, 2016, in Elife, where *agl62* mutant seeds have higher auxin activity than WT ones. The differences between the two studies could be due to the use of different reporters, DR5 and DII. But this should be cleared up, or at least discussed in the manuscript.

*1.2 Thank you for bringing up this important point. We used the same R2D2 sensor published by Liao et al. (2015). Previously, we did not show mDII:Tdtomato as the Tdtomato fluorescent signal was weak under Leica SP5 confocal. We now re-examined mDII:Tdtomato using a demo confocal (Leica Stellaris 8), which detected mDII:Tdtomato more easily. The new confocal data is added to the revised **Figure 3a**. The ratio between DII:VENUS and mDII:Tdtomato per endosperm nucleus was quantified and presented in **Figure 3a**. The data support that *atagl62* mutants have a reduced auxin in the endosperm. We do not know the reason behind the difference between our observation and that of Figueiredo et al, 2016, in Elife. We did use different reporters. We added a short discussion in the Discussion section (Line 512-520).*

3 - Page 13 – first paragraph – The authors claim that the precocious endosperm cellularization characteristic of the Arabidopsis *agl62* mutant can be rescued by exogenous application of auxin. I do not think there is enough evidence for this,

especially given the small sample sizes: around 50 seeds. In particular and given that the authors had to work with heterozygous plants, because *agl62* is zygotic lethal, only 25% of seeds per silique are homozygous *agl62*^{-/-}. Hence, within the same silique, how did the authors distinguish between a WT seed (genotypically *AGL62/AGL62* or *AGL62/agl62*) and an *agl62/agl62* seed with a rescued endosperm? These seeds should look the same.

*1.3 Thank you for the comments. *agl62*^{-/-} seeds are smaller in size (and eventually die) than the *AGL62* (+/-) and *AGL62* (+/+) seeds. Some auxin-rescued *agl62* (-/-) seeds with no cellularization is still smaller in size (see **Figure 3c**). However, it is possible to underestimate auxin-rescued *agl62*^{-/-} seeds if the seed size is restored.*

*In this revision, we quantified endosperm phenotype of all seeds derived from *AtAGL62*(+/-) parents irrespective of the seed size (**Figure 3d**) to avoid assumptions of genotypes based on seed size. Further, we examined a larger number of seeds, over 600 per treatment (**Figure 3d**). **In line 310-317 of the manuscript**, “We observed 22.7% (138 out of 608) mock-treated seeds exhibiting complete cellularization, a proportion that closely matches the 25% *atagl62-2* (-/-) seeds expected from the *atagl62-2* (-/+) parents”. “Upon 2,4-D treatment, only 15% (95 of 633) seeds exhibited complete cellularization (Fig. 3d)”. If one focuses on the homozygous *atagl62-2* (-/-) seeds, the proportion that exhibits complete cellularization would be reduced from 91% to 60% (22.7% and 15% divided by 25% respectively) by the 2,4-D treatment”.*

4 - Line 329 – I would be cautious with the conclusion that *FveAGL62/80* are not direct regulators of *FveYUC10*. The two transcription factors likely act as heterodimers. Therefore, an untargeted Y1H screen would likely not be sufficient to pick them up. Only actively expressing both AGLs in a yeast strain containing the *FveYUC10* promoter fragment would be sufficient to draw conclusions. I do not contest the data that *FveYUC10* is regulated by ATHBs, but would not rule out that *AGL62/80* could also regulate this gene directly.

*1.4 Thanks for the suggestion. We conducted a Yeast 1 hybrid assay using *FveAGL62* and *FveAGL80* alone or together as trans-acting factors. We tested both *FveYUC10* and *FveTAR1* promoters. We were unable to detect any trans-activation activity of *FveAGL62*, *FveAGL80*, alone or combination. This data is shown in **Supplementary Figure S9a** and described in Line 355-360.*

5 - Fig. 6 - It is clear that the overexpression of ATHBs leads to reduced fruit size in strawberry. However, no data is shown on whether seed development is also affected in these lines. If the authors want to make the point that ATHBs are regulators of auxin biosynthesis genes in the endosperm, then the overexpressor lines should produce seeds with the same defects as the *fveagl62* mutants. There is no indication that this is the case. There is also no evidence that ATHBs are expressed in the endosperm.

1.5a. Thank you for the suggestion. We examined the phenotype of ATHB-OE seeds under the confocal, which showed endosperm cell death as well as (but to a lesser extent) endosperm cellularization. The confocal images are added in **Figure 6f**, and the corresponding endosperm phenotype is quantified and shown in **Supplementary Figure S12**. Furthermore, we showed that auxin application increased the percentage of normal seeds (**Supplementary Figure S12**).

1.5b Regarding ATHB expression, we extracted RNA-seq data from our prior experiments to show ATHB expression in seeds (**Figure 5b**). The stage 1 RNA is from ovules, stage 2 RNA is from seeds, and stage 3 RNA is from Ghost (endosperm and seedcoat). This RNA-seq dataset was based on manual dissection of the seeds to the best of our ability. As shown in **Figure 5b**, the expression of ATHBs progressively goes down from stage 1 to stage 3, consistent with the repression of ATHB expression by AGL62/AGL80 in seeds. We apologize for failing to provide tissue details in **Figure 5b** in our previous manuscript.

6 - Page 16 – the claim that the ATHBs function downstream of AGL62/80 to regulate auxin biosynthesis is not fully tested. In particular, the authors do not show whether auxin activity as measured by DR5 is affected in the seeds of ATHB over-expression lines. Nor whether application of auxin can rescue the ATHB overexpressor phenotype, like it does to the *fveagl62* mutant. Furthermore, there is no evidence for a direct link between AGL62, ATHB and auxin production. I understand that this would require knocking-out ATHBs in the *fveagl62* mutant background, and that this is no easy task in strawberry. However, I do not think the authors have enough data to back up their claim of a regulatory module consisting of AGL62/80, ATHBs and auxin biosynthesis.

1.6a. Because of the time needed to cross or transform DR5 into the ATHB-OE transgenic plants (9 months to transform and 8 months to cross DR5 into ATHB-OE), we resorted to RT-qPCR, which showed a reduction of auxin biosynthesis gene expression in both ATHB29b-OE and ATHB30-OE lines (**Figure 6d-e**).

1.6b. Thank you for the suggestion. We added **Figure 6b**, which shows that the small fruit phenotype of ATHB-OE was rescued by the NAA application. Further, the NAA application reduces the percentage of seeds with dying endosperm cells and increases the percentage of normal seeds (**Supplementary Figure S12**).

1.6c. Due to the redundancy among ATHBs, we resorted to the over-expression of ATHBs to determine their function. In this revision, we provided ATHB-OE seed phenotype (**Figure 6f**) which resembles *fveagl62* endosperm phenotype in occasional early endosperm cellularization. More prominently, ATHB-OE seeds showed endosperm cell death (**Figure 6f**). The two phenotypes are quantified and shown in **Supplementary Figure S12**.

1.6d. To support the AGL62-ATHB module, we carried out the Y1H assay that confirmed a direct interaction between the AGL62/AGL80 dimer and the promoters of ATHB29b and ATHB30 (**Figure 5c, d**)(see Line 372-375).

Together, these new data have strengthened the AGL62-ATHB-auxin biosynthesis module. We appreciate very much all of your excellent suggestions.

Other points:

1 – Figure 1 – because the structure of the strawberry seed is not as known as the one from Arabidopsis, it would be helpful to have a legend indicating where the seed tissues are located. The authors mention that the GUS reporters are specifically expressed in the endosperm, but it is not clear to me where the endosperm starts and ends.

1.7 We added a schematic diagram of the strawberry seed (Figure 1c) that shows where the major seed tissues are located.

2 – The authors refer to strawberry seeds at stage 1 and 2, meaning pre- and post-fertilization. However, at stage 1 they should be referred to as “ovules” rather than “seeds”.

1.8 Thank you, we went through the manuscript to correct them.

3 – Page 10, first paragraph – DR5 expressed is used as a measure of auxin activity, not the actual amount of auxin. Therefore, “auxin level” should be corrected to “auxin activity”.

1.9 Corrected. (Line 224, 229)

4 – Line 226/227 – how many seeds were assayed in the mock-treated fveagl62 assay?

1.10 “n=56” fveagl62 seeds were mock-treated (Line 253).

5 – For the transactivation assays of Fig 5 and 6, what was used as an internal control, to normalize the LUC readings?

1.11 The Renilla luciferase (35S:REN) was used as a control to normalized the LUC readout. This information is added to the Method and Figure 5 legend.

6 – Figure S7 – The negative controls are missing for ATHB30.

1.12 We added a negative control testing the interaction between YFPc and YFPn-FveATHB30 (Supplementary Figure 10c).

7 – Which promoter was used for the overexpression of ATHBs in strawberry? This is not mentioned in the manuscript and I could not find information on the vector JH23 online.

1.13 The Arabidopsis UBQ10 promoter was used. This information is added to the Method and the text.

Reviewer #2 (Remarks to the Author):

The present manuscript by Guo et al on the " Mechanism of fertilization-induced auxin synthesis in the endosperm for seed and fruit development" attempts to demonstrate the mechanisms of how fertilization induces auxin biosynthesis in the endosperm to promote fleshy fruit initiation in strawberry. In the previous paper, Kang et al have reported that AtAGL62 plays an important role in the regulation of auxin synthesis in the Arabidopsis. Throughout the manuscript, several strawberry FveATHB genes were identified as downstream targets of FveAGL62 and act to repress auxin biosynthesis. This research topic itself is highly valuable for general readers of "Nature communication". However, it seems to me that there are some critical flaws with the manuscript. Biochemical and genetic evidence should be necessary to support the authors' claims. The following are some of the critical comments and experiments.

Thanks for the comments. The Kang et al Plant Cell (2008) reported the molecular cloning of AGL62 and AGL62' role in inhibiting endosperm cellularization. They did not investigate the role of AtAGL62 in the regulation of auxin synthesis. Our manuscript continues the functional dissection of AtAGL62 by revealing AtAGL62's role in promoting auxin biosynthesis in the endosperm.

1. Line111-112: Three mutant strains of FveAGL62 were obtained, but only one was selected in the follow-up experiment. I wonder why the other two mutants don't do follow-up experiments? If the phenotypes of the three mutant lines are consistent, wouldn't it be more illustrative?

*2.1 Thanks for the suggestion. **Supplementary Figure S3** is added to show the fruit phenotype as well as the endosperm early cellularization phenotype of two additional fveagl62 mutant lines, fveagl62^{-1/-1} and fveagl^{-1/-5}. Both lines showed lack of fruit enlargement and precocious endosperm cellularization identical to line 1 (fveagl62^{-1/-4}).*

2. Line138-140: Authors carried out Y2H and BiFC assays which showed that FveAGL62 and FveAGL80 interacted with each other. I think that author should provide more experimental evidence, for example, Co-IP or Pull-down.

*2.2 The interaction between AGL62 and AGL80 are well documented in the literature (de Folter S, et al. 2005 Plant Cell; Kang et al., 2008 Plant Cell). In this revision, we added additional BiFC data showing the positive interactions between FveAGL62 and all four FveAGL80-like proteins (FveAGL80L1, FveAGL80L2, FveAGL80L3, and FveAGL80L4) (**Supplementary Figure S4**). We hope that the Y2H (**Supplementary Fig. S2d**) and BiFC (**Supplementary Fig. S2e**), the additional interaction data between FveAGL62 and the four FveAGL80Ls (**Supplementary Figure S4**), and prior literatures provide reasonably strong supports for the interaction between AGL62 and AGL80. The Co-IP or pull-down need additional time and effort that we lack due to the large number of experiments we performed for the revision.*

3. Line141-143: These data suggest that FveAGL62 and FveAGL80 proteins may act together as heterodimers similar to their Arabidopsis homologs. They constructed fveagl62-1/-4, fveagl80-1/-1 double mutants, which exhibited a similar phenotype as the fveagl62-1/-4 single mutant. But, the fveagl80 -1/-1 mutant exhibited normal vegetative growth and developed fruit similarly to wild type (Figure S2C). I wonder whether the heterodimer of two proteins leads to the emergence of double mutant phenotype, or whether fveagl62-1/-4 gene knockout is the main factor. I think that experimental evidence supporting this conclusion is too weak.

2.3 I hope we understand your comments correctly. We provide a detailed explanation below:

*First, it is likely that heterodimers of **FveAGL62/FveAGL80** and **FveAGL62/FveAGL80L** are functionally similar (and redundant) to each other. Only when both types of heterodimers are abolished, could there be a mutant phenotype. In fveagl62 single mutants, neither FveAGL62/FveAGL80 nor FveAGL62/FveAGL80L heterodimers could form due to a loss of FveAGL62, hence the single fveagl62 mutants exhibit the seed and fruit phenotypes.*

*Second, FveAGL80 and FveAGL80Ls (AGL80L1, 2, 3, 4) can each form heterodimers with FveAGL62 (as shown in **Supplementary Fig. S2 and S4**). Therefore, in fveagl80 single mutants, heterodimers consisting of **AGL62/AGL80Ls** still form and offer the same function as FveAGL62/FveAGL80. Therefore, fveagl80 single mutants have no phenotype.*

Third, in fveagl62; fveagl80 double mutants, neither FveAGL62/FveAGL80, nor FveAGL62/FveAGL80L heterodimers can form, hence, the fveagl62; fveagl80 double mutants have the same phenotype as the fveagl62 single mutants which also lack both types of heterodimers.

*By adding the interaction assay between AGL62 and the four AGL80Ls (**Supplementary Figure S4**), we hope that you will find the single and double mutant phenotypes more comprehensible.*

4. Line172-176: The author identified six genes related to auxin synthesis by RNA-seq data, but the author only selected FveTAR1 and FveYUC10 for qRT-PCR verification. Why did the author only select two genes FveTAR1 and FveYUC10.

*2.4 We selected the two most highly expressed genes TAR1 and YUC10 for the RT-qPCR analysis. Per your request, we performed RT-qPCR data for three additional auxin biosynthesis genes, YUC5, YUC11, and TAA1, and have included this data in **Fig. 2c**.*

5. Line181-185: They found that the transcripts of many auxin and GA biosynthesis genes exhibited a gradual increase from stage 1 to stage 3 in the seeds, consistent with

their roles in fertilization-induced auxin biosynthesis in seeds. I think that authors should verify the expression of these genes in seeds by qRT-PCR.

*2.5 Based on your suggestion, we conducted RT-qPCR of **five** auxin biosynthesis genes and **four** GA biosynthesis genes in stage 1, 2 and 3 seeds. The data show a gradual increase in their transcript levels from stage 1 to 3 (**Supplementary Figure S6c**).*

6. Line329-331: 1) The Y1H experiment described in the results confirms that FveAGL62 or FveAGL80 cannot bind FveYUC10 directly, suggesting an indirect effect of FveAGL62 on auxin biosynthesis. However, there is no corresponding figures. The author should put the result figures of Y1H experiment in the manuscript or supplement data. 2) In addition to FveYUC10 gene, the author should use Y1H experiment to verify the relationship between FveAGL62 and other auxin synthesis genes identified in this paper.

*2.6 Thanks for the suggestion. We added Y1H data testing binding of FveAGL62 and FveAGL80 to the promoters of two auxin biosynthesis genes FveYUC10 and FveTAR1 (**Supplementary Figure 9a**). No binding was detected even when FveAGL62 and FveAGL80 were introduced into the yeast at the same time.*

7. Line 333-335: The authors introduction that one member of this subfamily was previously found to bind the FveYUC10 promoter in a Y1H screen. This experimental result is found in this manuscript or reported in previous studies. If it is previous research, references should be added.

*2.7 The Y1H screen was performed by us, in which we identified an ATHB member, FveATHB22 (FvH4_1g20040), as a transactivating factor of FveYUC10. This data is now added to **Supplementary Figure S9b**. Since this FveATHB22 is not highly expressed in the seed in vivo (Figure 5a), we did not pursue this gene. Instead, we pursued FveATHB29b and FveATHB30 as they both have a high level of expression in the seeds and show an elevated expression in the fveagl62 mutants.*

8. Line 341-346: The author only uses LUC assay to prove that FveAGL62 / FveAGL80 directly represses the expression of these FveATHBs, and there is too little experimental evidence. The authors should supplement EMSA or Y1H experiments to prove that FveAGL62/FveAGL80 can bind the promoter of FveATHBs and repress their expression.

*2.8 Per your suggestion, we performed Y1H assay testing the binding of FveAGL62/FveAGL80 to the promoters of ATHB29b and ATHB30. While single gene FveAGL62 or FveAGL80 failed to activate the reporter gene from the ATHB promoters, FveAGL62 and FveAGL80 together activated reporter expression from the FveATHB29b and FveATHB30 promoters, respectively (see **Fig. 5c, d** and Line 372-375).*

9. Line 387-391: In this paper, the author obtained the overexpression line of FveATHBs. It was proved that AGL62/AGL80 and ATHB to regulate auxin biosynthesis in the endosperm of fertilized seeds by measuring the size of transgenic fruit and the expression level of auxin synthesis related genes. I suggest that the phenotype of endosperm cellularization should be supplemented in FveATHB-OE.

*2.9 This is now provided in **Figure 6f** and **Supplementary Figure S12**. This is also suggested by other reviewers and explained in more detail in our response in **1.5a**.*

10. In general, the author only used knockout lines to study the function of AGL2 gene, while FveATHBs gene only used overexpression lines. If the article supplemented overexpression and knockout lines, the experimental data would be more sufficient.

2.10a Thank you for the suggestion. Strawberry transformation is a long and labor-intensive process that takes at least 9 months. We did try to over-express FveAGL62 in the past, but were unable to generate any positive transgenic plants. Since the problem appears specific to the FveAGL62 gene, we reasoned that over-expressing FveAGL62 might be toxic to the callus.

2.10b Since a large number of FveATHB genes both show increased expression in fveagl62 mutants and exhibit similar downward expression trends from stage 1 to stage 3 in WT seeds, we suspect that these FveATHB genes are functionally redundant. Therefore, knocking out multiple FveATHB family members will be necessary to reveal a loss-of-function phenotype. However, such a multi-knockout approach requires a highly efficient CRISPR system that is currently not feasible in strawberry. Therefore, we resort to the over-expression of FveATHBs to investigate the function of this subfamily of ATHB genes.

Reviewer #3 (Remarks to the Author):

The manuscript entitled “Mechanism of fertilization-induced auxin synthesis in the endosperm for seed and fruit development” reports a novel regulatory pathway for auxin induction in the endosperm after fertilization. In this work, authors show us some interesting findings regarding to auxin biosynthesis in strawberry and Arabidopsis endosperm. (1) Auxin synthesis is induced in the endosperm after fertilization. (2) MADS box gene AGL62 plays a critical role in the activation of auxin synthesis via AGL62-ATHB pathway.

Although the role of AGL62 in suppressing endosperm cellularization has been reported in Arabidopsis, its role in regulating endosperm development and the conservation of this mechanism among different species, as well as the downstream pathways of AGL62 remain largely unknown. This work provides novel knowledge of a fertilization triggered mechanism for activation of auxin synthesis in the endosperm. Together with previous works about Arabidopsis AGL62, this newly added information will enhance our understanding about endosperm development in angiosperms, especially about molecular linkage between fertilization and endosperm initiation. The

researchers in the field of plant development and horticulture will be interested in this work. Therefore, it is worthy to be known by readers.

Here, I list several concerns that may help authors to improve the manuscript before publication:

1. In Arabidopsis, previous work demonstrated that auxin response increases in the ovules after fertilization due to up-regulated auxin biosynthesis in the integuments (Robert et al., Nature Plants, 2018). Here, authors revealed that fertilization induces auxin biosynthesis in the endosperm. Some additional discussion in the manuscript may help readers to understand sequential of the auxin biosynthesis activation in different seed tissues in relation to fertilization.

*3.1 Thank you for the suggestion. Prior publications in Arabidopsis established that separate PRC2 complexes independently repress auxin synthesis in the seedcoat and the central cell before fertilization. Upon fertilization or pollination, the repression by the PRC2 is relieved, allowing seedcoat, embryo, and endosperm development. Auxin has been found to be synthesized in the endosperm, but its synthesis may not be exclusive to the endosperm. Per your request, we add discussion about auxin synthesis activation in different seed and floral tissues and cited **Robert et al (2018)** which shows fertilization-induced auxin synthesis in the integuments. Please see the last paragraph of the Discussion.*

2. In the manuscript, authors state that fertilization of the central cell leads to increased auxin biosynthesis. I am not sure whether fertilization in the central cell is a required step for auxin biosynthesis. It is very interesting to know if fertilization is indeed a required step for auxin biosynthesis activation in the endosperm. How about auxin biosynthesis activation in the autonomously developed endosperm as that in PRC2 mutants? What is the real trigger for auxin biosynthesis activation?

*3.2 Fertilization in the central cell is **not** a required step for auxin synthesis in mutants of PRC2. Figueiredo et al., (2015) showed that Arabidopsis mutants of gametophytic FIS2-PRC2 complex can induce ectopic auxin synthesis in the central cell and lead to the autonomous endosperm proliferation **in the absence of fertilization**. Further, AtAGL62 was shown a direct target of FIS2-PRC2 (Hehenberger et al., 2012) and atagl62 mutation can block autonomous endosperm development cause by mutants of PRC2. There is more discussion about this in the Discussion (Line 485-494).*

To trigger auxin biosynthesis, fertilization needs to remove the repressive effect of FIS2-PRC2 on AtAGL62, which can then activate auxin synthesis. Epigenetic factors and paternal contributions are proposed to play a role in this fertilization-induced activation of AGL62, but the actual mechanism is unknown.

3. Authors demonstrates that auxin is responsible for the phenotype of endosperm in agl62 mutants. Are there any developmental defects in the endosperms of mutants with

auxin biosynthesis defects? Related evidences will further confirm the conclusion in the present manuscript. In addition, GA biosynthesis related genes were also downregulated in agl62 mutants. Does GA also play a critical role in endosperm development?

3.3a. Earlier studies in Arabidopsis (Figueiredo et al. 2015) showed that the Arabidopsis wei8 tar1 tar2 triple mutants exhibited severe endosperm proliferation defects, indicating the requirement of auxin in endosperm proliferation (see Line 485-488 of Discussion). In strawberry, we and collaborators showed that CRISPR-knockout of FveYUC10 did not cause any seed or fruit phenotype (Feng et al., 2019 JXB), which is likely due to redundancy among the YUC and TAA/TAR genes in F. vesca.

3.3b. We applied GA to fveagl62 seeds but did not observe any difference in the endosperm phenotype between mock-treated and GA-treated seeds. GA was reported to mediate endosperm cell expansion for seed germination (Sanchez-Montesino et al 2019) and seed coat development (Figueiredo et al. eLife, 2016), but, to our knowledge, is not known to regulate early endosperm development.

4. In Figure S2, authors show us that FveAGL62 could interact with FveAGL80 in yeast. Besides FveAGL80, there are another four FveAGL80-Like genes in the strawberry genome. Could FveAGL62 also interacts with other FveAGL80-Like genes?

3.4 BiFC is used to test interactions between FveAGL62 and the four FveAGL80Ls (FveAGL80L 1, 2, 3 and 4) and found that FveAGL62 positively interact with all four FveAGL80Ls (Supplementary Figure S4).

5. The relationship between AGL62 and ATHB, ATHB and auxin biosynthetic genes were investigated by yeast system or transient tobacco system. In vivo Chip-PCR will further confirm the relationship between AGL62 and its downstream genes.

3.5 We agree that in vivo ChIP-PCR will be ideal to support binding of FveAGL62 to the downstream target genes. So far, we are unable to perform this experiment as we need to generate transgenic plants containing tagged-FveAGL62 or anti-FveAGL62 antibody which will take a lot longer time than is allowed for this revision.

In this revision, we provided additional Y1H data supporting a direct binding of FveAGL62/FveAGL80 to the promoters of ATHB29b and ATHB30 (Fig. 5c and 5d).

6. In this work, the whole seeds, including seed coats, endosperm and embryo, but not endosperm only, were collected for RNA-seq and RT-qPCR. Seed coats and embryos may influence the analysis and interpretation of the results. Is it possible that auxin is synthesized in the seed coats and transports to the endosperm?

3.6. Due to limitations of our ability to manually separate the seedcoat from the endosperm for RNA extraction, we used reporter gene expression to show that

FveAGL62 and FveAGL80 are expressed specifically in the endosperm but not in the seedcoat (Figure 1a, b).

To examine auxin and GA biosynthesis genes by RT-qPCR (Figure 2c and Supplementary Figure S6), we isolated three tissues, stage 1 ovule, stage 2 seed, and stage 3 ghost (seedcoat and endosperm). For some of these genes, their endosperm-preferential expression (and an absence of seedcoat expression) was shown in an earlier study using the GUS reporter including TAR1, YUC5, and YUC11 (Feng et al. JXB 2019). Therefore, while we couldn't rule out auxin's synthesis in the seedcoat, the YUC-GUS and TAA/TAR1-GUS reporter expression does not support it.

7. Authors show us an important finding that Strawberry FveAGL62 could rescue Arabidopsis atagl62 mutant seeds. More detailed evidences such as how many transgenic lines show normal seed setting as wild type plants and detailed endosperm phenotype analysis in the FveAGL62 transgenic lines will underpin this new finding. In addition, comparative analysis of AGL62 genes in different representative plants during evolution will provide useful information about AGL62.

3.7a We generated 12 independent transgenic lines in Arabidopsis; all of the lines developed normal seeds. Using confocal microscopy, we analyzed three lines in depth. Specifically, we examined the rescue of early endosperm cellularization by the FveAGL62 transgene and showed almost 100% rescue of the endosperm phenotype by the transgene (Supplementary Figure S8).

3.7b Do you suggest evolutionary comparisons of agl62 mutant phenotypes in different plant species? In this study we compared Arabidopsis agl62 mutants with the strawberry agl62 mutants. However, we are unable to find any literature on the genetic (functional) studies of AGL62 in other plant species.

8. In the method section, the detailed information about FveATHB29b and FveATHB30 overexpression is missing. Which promoter was used to drive FveATHB29b and FveATHB30 overexpression in the endosperm? Is it exclusively active in the endosperm?

3.8 The promoter used to over-express FveATHBs is the Arabidopsis Ubiquitin (UBQ) 10 promoter. Hence, the over-expression is not exclusive to the endosperm. This information is added to the Method section.

9. In Figure 6, only the phenotype of fruits was analyzed in FveATHB29b-OE and FveATHB30-OE transgenic lines. Detailed phenotype analysis of the endosperm in FveATHB29b and FveATHB30 overexpression plants will help to confirm the AGL62-ATHB29 pathway.

3.9 *Thank you. Confocal images of FveATHB29b-OE and FveATHB30-OE transgenic seeds are added to **Figure 6f**. Quantitative analysis of their endosperm phenotype is presented in **Supplementary Figure S12**.*

10. In Figure 6, authors demonstrated that auxin biosynthesis genes such as YUC10 and TAA1 were significantly down-regulated in FveATHB29b-OE and FveATHB30-OE transgenic plants. Since treatment of fveagl62 mutants with GA could also lead to fruit and seed enlargement, whether GA biosynthesis pathways were also affected in FveATHB29b and FveATHB30 overexpression lines?

3.10 *Thank you for the suggestion. We conducted RT-qPCR of four different GA biosynthesis genes in FveATHB-OE lines. While three of the GA biosynthesis genes (FveGA20ox1c, FveGA3ox1a, and FveGA3ox1b) showed a reduction of expression, one (FveGA20ox1d) showed an increased expression when compared with the non-transgenic plants. Since we are unable to make a definitive conclusion regarding how the GA biosynthesis is impacted in the FveATHB-OE lines, we did not include this RT-qPCR data in the manuscript. The role of GA in seed and fruit development requires further investigation.*

11. In the tobacco transient assay, FveATHB30, but not FveATHB29b, could repress the expression of ProFveYUC10::LUC. But the expression levels of YUC were significantly reduced in both FveATHB29b and FveATHB30 overexpression lines. These two results seem not in line with each other.

3.11 *ATHBs function through dimerizing with other family members (Tan and Irish, 2006), and our study showed that FveATHB29b and FveATHB30 could dimerize with each other (**Supplementary Figure S10**), and may each heterodimerize with other ATHB subfamily members in vivo. The heterodimers formed in the over-expressed lines may be the ones that execute the repression, and they may differ from the dimers formed in the transient expression assay in tobacco. We added a short explanation in Line 408-411.*

12. When comparing the fruit phenotype in FveATHB29b-OE and FveATHB30-OE transgenic lines, fruit development seems affected more seriously in FveATHB29b-OE. But the ability of ATHB30 in repressing the expression of downstream auxin biosynthetic genes, such as YUC10, is stronger than ATHB29b as shown in Figure 6b,e,f. Potential reasons for the differences could be discussed in the manuscript.

3.12 *See our explanation above (in 3.11). In addition, the FveATHB29b are over-expressed at a much higher level (10-20 fold) than that of FveATHB30 (4-8 fold) (Supplementary Fig. S11) in the OE transgenic lines. The much higher expression of FveATHB29b in FveATHB29b-OE lines may lead to a stronger phenotype than the FveATHB30-OE lines.*

REVIEWERS' COMMENTS

Reviewer #1 (Remarks to the Author):

I commend the authors for addressing all the concerns that I had raised. I have no further suggestions for improvements. This is a very nice and well-written story and I support its publication in Nature Communications.

Reviewer #2 (Remarks to the Author):

The mechanisms of how auxin biosynthesis in the endosperm impacts flight fruit initiation in strawberries are described in more detail. The experimental evidence is relatively complete. Although they have provided more detailed results to support their conclusion, some minor questions they need to clarify before it is accepted.

1. In strawberry Gene IDs part: all genes should be italicized
2. The materials and methods need to be revised. More detailed information about should be added.

Reviewer #3 (Remarks to the Author):

I have evaluated the manuscript before and think that the authors have essentially improved their interesting manuscript. Previously, I don't have any critical concern with their major conclusions and in this revised manuscript they have further clarified all the queries. Thus, I think the manuscript can be accepted as it is.

Dear reviewers,

Thank you so much for your time in re-reviewing our revised manuscript and for your positive comments. The point-by-point response to your comments are below:

Reviewer #1 (Remarks to the Author):

I commend the authors for addressing all the concerns that I had raised. I have no further suggestions for improvements. This is a very nice and well-written story and I support its publication in Nature Communications.

Thank you so much for your kind encouragement and recommendation.

Reviewer #2 (Remarks to the Author):

The mechanisms of how auxin biosynthesis in the endosperm impacts flight fruit initiation in strawberries are described in more detail. The experimental evidence is relatively complete. Although they have provided more detailed results to support their conclusion, some minor questions they need to clarify before it is accepted.

Thank you so much for your kind encouragement and recommendation.

1. In strawberry Gene IDs part: all genes should be italicized

Thank you for the suggestion. The genes are italicized now. (Lines 486-495)

2. The materials and methods need to revised. More detailed information about should be added.

Thank you for the suggestion. More detailed information on agrobacterium transformation of strawberry calli, seed fixation for confocal microscopy, and agrobacterium infiltration of tobacco leaves in transient gene expression analyses are added or/and expanded in the Methods. Track change is used to highlight these additions.

Reviewer #3 (Remarks to the Author):

I have evaluated the manuscript before and think that the authors have essentially improved their interesting manuscript. Previously, I don't have any critical concern with their major conclusions and in this revised manuscript they have further clarified all the queries. Thus, I think the manuscript can be accepted as it is.

Thank you so much for your kind encouragement and recommendation.